# Cd34+ Stromal Cells/Telocytes in Normal and Pathological Skin

**DOI:** 10.3390/ijms22147342

**Published:** 2021-07-08

**Authors:** Lucio Díaz-Flores, Ricardo Gutiérrez, Maria Pino García, Miriam González-Gómez, Rosa Rodríguez-Rodriguez, Nieves Hernández-León, Lucio Díaz-Flores, José Luís Carrasco

**Affiliations:** 1Department of Basic Medical Sciences, Faculty of Medicine, University of La Laguna, 38071 Tenerife, Spain; histologia54@gmail.com (R.G.); mirgon@ull.es (M.G.-G.); rosyroguez@gmail.com (R.R.-R.); nievleon34@gmail.com (N.H.-L.); ldfvmri@yahoo.com (L.D.-F.J.); jcarraju@ull.edu.es (J.L.C.); 2Department of Pathology, Eurofins Megalab–Hospiten Hospitals, 38100 Tenerife, Spain; mpgarcias@megalab.es

**Keywords:** stromal cells, telocytes, skin histology, non-tumoral skin pathology, skin tumors, CD34

## Abstract

We studied CD34+ stromal cells/telocytes (CD34+SCs/TCs) in pathologic skin, after briefly examining them in normal conditions. We confirm previous studies by other authors in the normal dermis regarding CD34+SC/TC characteristics and distribution around vessels, nerves and cutaneous annexes, highlighting their practical absence in the papillary dermis and presence in the bulge region of perifollicular groups of very small CD34+ stromal cells. In non-tumoral skin pathology, we studied examples of the principal histologic patterns in which CD34+SCs/TCs have (1) a fundamental pathophysiological role, including (a) fibrosing/sclerosing diseases, such as systemic sclerosis, with loss of CD34+SCs/TCs and presence of stromal cells co-expressing CD34 and αSMA, and (b) metabolic degenerative processes, including basophilic degeneration of collagen, with stromal cells/telocytes in close association with degenerative fibrils, and cutaneous myxoid cysts with spindle-shaped, stellate and bulky vacuolated CD34+ stromal cells, and (2) a secondary reactive role, encompassing dermatitis—e.g., interface (erythema multiforme), acantholytic (pemphigus, Hailey–Hailey disease), lichenoid (lichen planus), subepidermal vesicular (bullous pemphigoid), psoriasiform (psoriasis), granulomatous (granuloma annulare)—vasculitis (leukocytoclastic and lymphocytic vasculitis), folliculitis, perifolliculitis and inflammation of the sweat and sebaceous glands (perifolliculitis and rosacea) and infectious dermatitis (verruca vulgaris). In skin tumor and tumor-like conditions, we studied examples of those in which CD34+ stromal cells are (1) the neoplastic component (dermatofibrosarcoma protuberans, sclerotic fibroma and solitary fibrous tumor), (2) a neoplastic component with varying presentation (fibroepithelial polyp and superficial myxofibrosarcoma) and (3) a reactive component in other tumor/tumor-like cell lines, such as those deriving from vessel periendothelial cells (myopericytoma), epithelial cells (trichoepithelioma, nevus sebaceous of Jadassohn and seborrheic keratosis), Merkel cells (Merkel cell carcinoma), melanocytes (dermal melanocytic nevi) and Schwann cells (neurofibroma and granular cell tumor).

## 1. Introduction

CD34+ stromal cells/telocytes (CD34+SCs/TCs) are an important interstitial, perivascular, peri/endoneurial and periadnexal cellular component in the dermis and hypodermis (subcutaneous adipose tissue) of the skin. Most CD34+ stromal cells are known to correspond to telocytes, a new cellular type identified by electron microscopy [1,2]. Ultrastructurally, these cells show a small somatic body and two to five long cytoplasmic processes (telopodes) and are located in the stroma of multiple anatomical sites. Although telocytes show a characteristic ultrastructure, they present immunophenotypic heterogeneity, depending on anatomical location [3]. In addition to CD34 positivity TCs are also described as CD34/PDGFRα double positive [4,5,6,7,8,9,10,11]. Expression of vimentin, CD117, CD29, VEGF, and estrogen and progesterone receptors has also been observed in some locations [12,13,14,15,16,17]. The immunophenotype profile of telocytes (CD34+/PDGFRα+/vimentin+/CD31-) differentiates them from fibroblasts (CD34-/PDGFRα+/vimentin+/CD31-) and endothelial cells (CD34+/PDGFRα-/vimentin+/CD31+) [18]. In general, a specific immunomarker for TCs is still an issue of study. Several roles have been hypothesized for CD34+SCs/TCs, including intercellular communication, control and organization of the extracellular matrix, structural support, endocytosis, creation of tissular microenvironments, guidance to migration, contribution of scaffolds, immunomodulation, neurotransmission, control and regulation of other cell types, stem cell modulation and participation in regeneration, repair and tumor stroma formation [1,2,3,13,15,19,20,21,22,23,24,25,26,27,28,29,30,31]. In addition, several tumors can have their origin in CD34+SCs/TCs [32,33,34]. In the skin, the suggested roles of CD34+SCs/TCs include mechanical support, regeneration (tandem between TCs and stem cells), communication (intercellular contacts and extracellular vesicles) and endocytosis, immune regulation, modulation of fibroblasts, mast cells and macrophages, reduction of inflammatory response, participation in metabolism, homeostasis, (neo) angiogenesis and in the interaction between collagen and elastic fibers [3,4,13,14,15,18,22,35,36,37,38,39,40,41,42,43,44,45].

Our objective is to review the characteristics and morphologic behaviour of CD34+SCs/TCs in normal and pathological skin (Table 1), contributing our own observations. Because CD34+SCs/TCs are interstitial cells, they can be part of, related to or influenced by pathologic processes of the various anatomical components present in the skin. These anatomical components of the skin comprise the epidermis, the melanocytic system, the dermal connective tissue and individual cells (macrophages, mast cells and dermal Langerhans cells), the microvasculature, preterminal cutaneous nerves, sensory nerve endings and Merkel cells, eccrine and apocrine sweat glands, hair follicles, sebaceous glands, adipose tissue and the nail. To obtain a broad view of CD34+SCs/TCs in skin pathology, we will explore the most important histological patterns in dermatopathology [46], by providing examples. In previous works, we studied these cells in the peripheral nervous system [32] and in adipose tissue [33], both in normal and pathologic conditions. Therefore, CD34+SCs/TCs will only be briefly considered in these skin components.

## 2. CD34+SCs/TCs in the Normal Skin

Several authors have studied the behavior of telocytes in the normal skin, establishing their characteristics and distribution [13,35,36,37,38,39,40,41,42,43,44]. We will briefly consider these aspects of CD34+SCs/TCs in the skin, especially their dermal location and arrangement in the interstitium and around blood and lymphatic vessels, nerves and cutaneous annexes (hair follicles, and sweat and sebaceous glands).

CD34+SCs/TCs are more numerous in the reticular and deep dermis than in the superficial dermal layer (Figure 1A). These CD34+ cells show similar morphologic characteristics to CD34+SCs/TCs in other tissues and organs, with a small somatic region and bipolar or multipolar, long, thin processes (telopodes) (Figure 1B–E). In the interstitium, CD34+SCs/TCs are arranged among collagen bundles and elastic fibers.

The distribution of CD34+SCs/TCs around blood and lymphatic vessels in the skin varies depending on the size, location and type of vessel. All skin vessels, except those in the papillary dermis, are surrounded by CD34+SCs/TCs, which are an important component in the adventitia of the larger vessels and form a delimiting layer in most of the smaller ones (Figure 1F,G). If we consider this delimiting layer of the smallest vessels as a very thin adventitia, we can posit that CD34+SFs/TCs extend continuously through most of the skin blood vessels and constitute an important component of their adventitia. As in other locations, these CD34+SCs/TCs show long, thin processes (Figure 1F,G), with ultrastructural characteristics of telopodes (Figure 1H). Regarding vessel location in the skin [51,52,53], we will consider the CD34+SCs/TCs in vessel loops in the papillary dermis, the two horizontal vascular plexuses, including the upper, located 1–1.5 mm below the skin surface, and that situated at the dermal subcutaneous interface, as well as the vessels that connect both plexuses. The papillary vessel loops, formed by ascending precapillary arterioles–capillaries and descending capillary–postcapillary venules that originate and terminate, respectively, in the upper horizontal plexus show endothelial and mural cells (vascular smooth muscle cells and pericytes), but are devoid of CD34+SCs/TCs (Figure 2A,B). This important finding also occurs in the mucosa layer of some organs (e.g., intestine and gallbladder) [8,54] and can explain the behavior of some pathological processes (see below). Except in the papillary dermis, CD34+SCs/TCs are found around vessels and in the interstitial tissue in the dermis, including the upper horizontal plexus (Figure 2C) and the dermal subcutaneous junction plexus. The arteries in the dermal subcutaneous junction plexus show several layers of CD34+SCs/TCs (Figure 2D). The veins in this plexus, equipped with two cusped valves (Figure 2E) that prevent the retrograde flow of blood [51], also present CD34+SCs/TCs, though fewer than in the arteries (Figure 2E). In addition, CD34+SCs/TCs are observed surrounding small groups of smooth muscle cells in the pre-collector and collector lymphatic vessels in the dermal subcutaneous junction (Figure 2F,G).

CD34+SCs/TCs are also concentrated around skin annexes, including the arrector pili muscle (hair erector muscles) (Figure 2H). CD34+SCs/TCs are prominent around the excretory, ductal and secretory portions of the sweat glands, originating intertwined networks (Figure 3A,B). In hair follicles, CD34+SCs/TCs extend from the infundibulum to the deep segments, forming a varying number of parallel layers (Figure 3C,D). Sebaceous glands are also surrounded by typical CD34+SCs/TCs, with long, thin processes (Figure 3E).

We observed isolated clusters of very small, closely grouped CD34+ stromal cells with multiple intricate processes in the stroma around the hair follicle between the opening of the sebaceous gland and the end of the erector muscle attachment (bulge region) (Figure 3F). Clusters were separated from hair and the sebaceous gland epithelium by rows of CD34+SCs/TCs (Figure 3F). The very small CD34+ stromal cells (minute/dwarf CD34+SCs/TCs?) may correspond to the mesenchymal stromal/stem cells isolated from the hair follicle dermal sheath [55], an issue that requires further studies.

## 3. CD34+SCs in Non-Tumoral Pathological Skin

The behavior of CD34+SCs/TCs in most non-tumoral diseases of the skin has received little attention. Exceptions are psoriasis, scleroderma (systemic sclerosis), and wound healing on which several studies have been published [38,45,47,48,49,56,57,58,59,60]. We will first consider the non-tumoral pathological processes in which CD34+SCs/TCs have important pathophysiological implications, mainly related to the role of CD34+SCs/TCs in tissue homeostasis, control and organization of the extracellular matrix, and regeneration and repair. These processes can be grouped in (a) fibrosing/sclerosing diseases, such as morphea, systemic sclerosis (scleroderma), lichen sclerosus and radiation dermatitis, (b) metabolic and degenerative processes of the dermal extracellular matrix, including basophilic degeneration of the dermis/collagen, cutaneous amyloidosis and dermal mucinosis and (c) wound healing and formation of capsules (encapsulation phenomena). In a second section we will examine the non-tumoral processes in which CD34+SCs have a secondary reactive role. Given the considerable scope of this last section, we have grouped the different non-tumoral skin diseases by histological patterns and succinctly present CD34+SC/TC behavior in one or two examples of each.

### 3.1. Non-Tumoral Pathologic Processes of the Skin in Which CD34+SCs/TCs Have a Fundamental Pathophysiologic Role

We will describe the characteristics and evolution of CD34+SCs/TCs in examples of this section, including systemic sclerosis (scleroderma) in fibrosing/sclerosing diseases, basophilic degeneration of the collagen and cutaneous myxoid cysts in metabolic and degenerative disorders of the dermal extracellular matrix. We have already reviewed an example of CD34+SCs/TCs in skin amyloidosis in a previous work [33] and as an example of repair will present the peritumoral fibrosis and encapsulation in the Section 4.3.1.

#### 3.1.1. CD34+SCs/TCs in Fibrosing/Sclerosing Processes (Systemic Sclerosis)

In systemic sclerosis (scleroderma), in which a derangement of the microvascular system occurs [61], a progressive reduction and loss of CD34+SCs/TCs has been described in the dermal cellular network [38]. This phenomenon also occurs in different tissues/organs affected by this process [47], with the appearance of a varying number of myofibroblasts [56,59]. The possible source of myofibroblasts in scleroderma was also considered [56,59], as well as the potential dual role (anti-fibrotic and pro-fibrotic) of adipose-derived stromal cells in systemic sclerosis [59]. In accordance with the aforementioned descriptions by other authors, we have observed loss of CD34+SCs/TCs and presence of myofibroblasts in scleroderma (Figure 4A,B). In some regions CD34+SCs/TCs were present around collagen fibers expressing Collagen I (Figure 4C), while in other regions these cells were greatly diminished in number (Figure 4D). Using double immunofluorescence labelling for CD34 and αSMA, we observed coexpression of both markers in stromal cells of scleroderma (Figure 4E), which supports the notion that resident CD34+SCs/TCs may be a source of myofibroblasts in this lesion. This issue requires further studies.

#### 3.1.2. CD34+SCs/TCs in Metabolic and Degenerative Processes of the Dermal Extracellular Matrix

In this section we explore the basophilic degeneration of collagen and cutaneous mucinosis. The latter comprises numerous processes classified as diffuse and focal primary mucinosis, as well as focal, tumor-like mucinosis. We have chosen the cutaneous myxoid cyst as an example of these processes.

##### CD34+SCs/TCs in Basophilic Degeneration of Collagen

In basophilic degeneration of collagen (senile/actinic atrophy of the skin), CD34+SCs/TCs are observed around collagen and elastic dermal components with degenerative changes. Ultrastructurally, the long, moniliform processes of telocytes, showing podomeres and podons, are observed in close association with several degenerative fibers in the dermis (Figure 4F,G).

##### CD34+SCs/TCs in Cutaneous Mucinosis (Cutaneous Myxoid Cysts)

Cutaneous myxoid cysts [62] originating away from joints are examples of focal CD34+SCs/TCs with an increased production capacity of acid mucopolysaccharides (hyaluronic acid). These cells are arranged in the loose connective tissue of the dermis, in which varying size spaces are devoid of cells (myxoid lagoons) (Figure 5A,B). CD34+SCs/TCs show a spindle-shaped, stellate or irregular morphology with long, moniliform processes (Figure 5C,D). Bulky, multi-vacuolated CD34+ mononuclear cells are often observed (Figure 5E,F). These voluminous cells of increased somatic size show some retracted processes and resemble modified CD34+SCs/TCs in the plexiform type of neurofibroma, in which myxoid deposits can be prominent [32]. Myxoid lagoons are partially or totally covered by CD34+SCs/TCs (Figure 5A–C), as occurs around spaces with abundant extracellular glycosaminoglycans (hyaluronic acid) [63].

### 3.2. Non-Tumoral Pathologic Processes of the Skin in Which CD34+SCs/TCs Have a Secondary Reactive Role

In this section, we will examine the behavior of reactive CD34+SCs/TCs in examples of the pathological processes included in the following non-tumoral histopathologic patterns of the skin [46]: interface dermatitis (erythema multiforme), acantholytic dermatosis (pemphigus and Hailey–Hailey disease), lichenoid dermatitis (lichen planus), subepidermal vesicular dermatitis (bullous pemphigoid), psoriasiform dermatitis (psoriasis), granulomatous dermatitis (granuloma annulare), vasculitis (leukocytoclastic and lymphocytic vasculitis), folliculitis, perifolliculitis and inflammation of the sweat and sebaceous glands (folliculitis and rosacea), and infectious dermatitis (verruca vulgaris).

#### 3.2.1. CD34+SCs/TCs in Interface Dermatitis. Erythema Multiforme

Erythema multiforme is an important example of interface dermatitis, which also includes other processes, such as lupus erythematosus, toxic epidermal necrolysis, dermatomyositis, interface dermatitis of HIV infection, pityriasis lichenoid and graft-versus-host disease.

In early lesions of erythema multiforme, with hydropic degeneration of basal cells, cytoid bodies and lymphocytic infiltration (Figure 6A), interstitial and perivascular CD34+SCs/TCs are not observed in the inflammatory infiltrate in superficial dermal areas (Figure 6B). However, CD34+SCs/TCs surround the occasional, scant perivascular lymphocytic infiltrate present in the vessels of other dermal layers. In lesional areas with vesiculation or blister formation, varying numbers of perivascular and interstitial CD34+SCs/TCs are observed in the underlying dermis (Figure 6C,D).

#### 3.2.2. CD34+SCs/TCs in Acantholytic Dermatosis. Pemphigus. Hailey–Hailey Disease

##### CD34+SCs/TCs in Pemphigus

The characteristics of CD34+SCs/TCs in the varying superficial, perivascular lymphocytic inflammatory infiltrate, often along with eosinophils and some neutrophils, were similar in the different types of pemphigus. Generally, perivascular CD34+SCs/TCs were seen surrounding the inflammatory cells grouped around the pericytes or vascular smooth muscle cells (Figure 6E), with greater evidence in the deep portions of the inflammatory infiltrate. Conversely, some of the most superficial vessels showed no surrounding CD34+SCs/TCs (Figure 6F).

##### CD34+SCs/TCs in Hailey–Hailey Disease

In Hailey–Hailey disease, which encompasses papillomatosis, acanthosis and acantholysis, and has a characteristic image resembling a dilapidated brick wall (Figure 7A,B), CD34+SCs/TCs are absent in the more superficial portion of the moderate subepidermal inflammatory infiltrate present in this lesion (Figure 7C). Conversely, in the deeper portion, interstitial and perivascular CD34+SCs/TCs are observed surrounding small aggregates of mononuclear cells (Figure 7D).

#### 3.2.3. CD34+SCs/TCs in Lichenoid Dermatitis. Lichen Planus

Lichen planus is the main example of dermatitis with a lichenoid pattern, including the lichenoid inflammatory infiltrate. In addition to lichen planus, this group of processes comprises lichen striatus, lichen nitidus, lichenoid drug reaction, lichenoid keratosis, fixed drug eruption and erythema dyschromicum perstans. The presence, characteristics and relationship with the vascularization and inflammatory infiltrate of CD34+SCs/TCs in lichen planus vary depending on the dermal layer. Thus, CD34+SCs/TCs are absent in the superficial dermis (Figure 7E,F), in which the characteristic band-like inflammatory infiltrate (lichenoid inflammatory infiltrate) is observed below and parallel to the epidermis. Vessels with open or closed lumens are seen between the chronic inflammatory infiltrate formed by lymphocytes and macrophages (plasma cells are generally absent). These vessels show endothelial and mural cells, but no surrounding CD34+SCs/TCs (Figure 7E,F). Where a collagenous area is present below the inflammatory infiltrate, few vessels are observed and there are no surrounding CD34+SCs/TCs (Figure 7G,H). The number and distribution of underlying CD34+SCs/TCs are within normal limits. Interestingly, CD34+SCs/TCs are observed around the inflammatory cells in vessels with perivascular inflammatory infiltrate in this underlying region (Figure 7I).

#### 3.2.4. CD34+SCs/TCs in Subepidermal Vesicular Dermatitis. Bullous Pemphigoid

In this section, which includes bullous pemphigoid, mucous pemphigoid, epidermolysis congenita and acquisita, dermatitis herpetiformis, pemphigoid gestationis (herpes gestationis), porphyria cutanea tarda, pseudoporphyria and linear IgA disease, we will use bullous pemphigoid as an example.

In the dermal connective tissue underlying the subepidermal blister of bullous pemphigoid, with a perivascular infiltrate of mononuclear cells and numerous eosinophils (Figure 8A), the presence of CD34+SCs/TCs ranges from scarce to moderate (Figure 8B–E). In areas where CD34+SCs/TCs are moderate in number, these cells are prominent with long processes (Figure 8C) surrounding the perivascular lymphohistiocytic infiltrate (Figure 8D,E).

#### 3.2.5. CD34+SCs/TCs in Psoriasiform Dermatitis. Psoriasis

Telocytes have been studied extensively in psoriasis by immunochemistry, immunofluorescence and transmission electron microscopy [49]. The most notable findings for the distribution and characteristics of CD34+SCs/TCs in psoriasis with elongated rete ridges (Figure 9A) are seen in the superficial and papillary dermis. Between the mild superficial perivascular inflammatory lymphocytic infiltrate, varying numbers of CD34+SCs/TCs are observed around vessels (Figure 9B,C). Perivascular CD34+SCs/TCs gradually decrease as the vessels ascend in the superficial layer (Figure 9B,D), and loss of perivascular and interstitial CD34+SCs/TCs is observed from the base to the apex of the papillary dermis (Figure 9B,D,E). Thus, the papillary dermis between the rete ridges—typically elongated in psoriasis—shows tortuous vessels that are dilated or have virtual or small lumens and are devoid of perivascular CD34+SCs/TCs (Figure 9F–I). Vessels with small lumens present interendothelial apical (Figure 9F) or planar (Figure 9G) contacts. Pericytes are observed in the folds of the tortuous vessels (Figure 9H). Occasionally, regressive phenomena are seen in intrapapillary vessels (Figure 9I).

#### 3.2.6. CD34+SCs/TCs in Granulomatous Dermatitis. Granuloma Annulare

CD34+SCs/TCs were not observed in granulomas of this group but were present in the surrounding dermis. Granuloma annulare, which shows degenerated collagen, variable mucin deposition, loss of elastic component and a palisading (Figure 10A,B) or interstitial inflammatory infiltrate (palisading and interstitial variants), is a good example of the CD34+SC/TC response in skin granulomatous lesions. Thus, CD34+SCs/TCs are observed around the palisading inflammatory infiltrate, including perivascular (Figure 10C) and interstitial (Figure 10D) locations. The vessels that penetrate the granuloma do not show perivascular CD34+SCs/TCs (Figure 10C). Small accumulations of lymphocytes are also observed in the spaces between the perivascular CD34+SCs/TCs and the vessel mural cells (pericytes or vascular smooth muscle cells) (Figure 10E).

#### 3.2.7. CD34+SCs/TCs in Vasculitis (Leukocytoclastic and Lymphocytic Vasculitis)

In leukocytoclastic vasculitis, with a perivascular neutrophilic infiltrate, neutrophil karyorrhexis, erythrocyte extravasation and some mononuclear cells (Figure 11A), the presence of CD34+SCs/TCs varies (Figure 11B,C), depending on the vessel location. CD34+SCs/TCs are not observed in the most superficial areas of the inflammatory infiltrate, which surrounds vessels with endothelial swelling and focal necrosis (Figure 11C). Conversely, CD34+SCs/TCs are present in other areas of the inflammatory infiltrate (Figure 11B). Debris of some CD34+SCs/TCs is also seen. In the heterogeneous group of lymphocytic vasculitis, CD34+SCs/TCs are present around the perivascular mixed inflammatory infiltrate (Figure 11D).

#### 3.2.8. CD34+SCs/TCs in Folliculitis, Perifolliculitis, Inflammation of the Sebaceous Glands (e.g., Perifolliculitis and Rosacea) and Infectious Diseases (Verruca Vulgaris)

CD34+SCs/TCs are absent in perifolliculitis (Figure 11E) and the perifollicular and perivascular inflammatory infiltrate present in rosacea (Figure 11F). They are, however, present around inflammatory infiltrates in both conditions. In rosacea, the absence of CD34+SCs/TCs is irrespective of the inflammatory cells in the different stages of the lesion (lymphocytic, mixed or granulomatous). CD34+SCs are also absent in the peribulbar lymphocytic infiltrate in alopecia areata. In verruca vulgaris, dermal areas can be observed with (Figure 11G) or without (Figure 11H) interstitial and perivascular CD34+SCs/TCs.

## 4. CD34+ Stromal Cells in Tumor/Tumor-Like Conditions of the Skin

In the tumor/tumor-like conditions of the skin containing CD34+ stromal cells, these cells can be the neoplastic component or secondary reactive stromal component of tumors of other cell lines. Likewise, when CD34+ stromal cells are a neoplastic component, CD34 expression may or may not be constant in all cases and may occur in all, or varying percentages, of stromal cells. Next, we give examples of these conditions.

### 4.1. Tumor/Tumor-Like Conditions of the Skin in Which CD34+ Stromal Cells Are the Neoplastic Component

In this section, we will study dermatofibrosarcoma protuberans, sclerotic fibroma (circumscribed storiform collagenoma) and solitary fibrous tumor, as examples of tumor/tumor-like conditions with expression of CD34 in stromal cells.

#### 4.1.1. CD34+ Stromal Cells in Dermatofibrosarcoma Protuberans

Dermatofibrosarcoma protuberans, a highly cellular, low-grade tumor, is a good example of skin neoplasms formed by cells of mesenchymal lineage that express CD34 [64,65,66], The CD34+ neoplastic cells are spindle-shaped (Figure 12A–C), show little pleomorphism and are arranged in interwoven fascicles, giving the lesion a storiform or intersecting (cartwheel) pattern (Figure 12A). These cells involve the dermis and subcutaneous adipose tissue (Figure 12D) (rare cases are limited to the dermis), occasionally reaching the fascia, muscle and periosteum. The epidermis is normal or thinned and is separated from the lesion by a thin strip of connective tissue. Vessels of varying size (Figure 12E), sweat glands (Figure 12F), nerves (Figure 12F) and erector hair muscle appear among numerous CD34+ neoplastic cells, which merge with those that normally surround these structures. In addition to CD34 expression, the neoplastic cells present positivity for vimentin. Conversely, αSMA, S-100 protein, melan-A, cytokeratin and epithelial membrane antigen are not expressed. These cells show long, thin bipolar processes (Figure 12G), whose nuclei are described as inconspicuous and elongated, with mild hyperchromasia and small nucleoli (see below). Mitotic activity is moderate, with no atypical mitoses, and the proliferative index (Ki 67) ranges between 1% and 22%. S-100+ pigmented cells can be associated with CD34+ cells (pigmented type, Bednar tumor), and several giant CD34+ cells can be seen in certain tumors (Giant cell type). CD34+ cells are arranged between a collagenous stroma in which Collagen I is observed. Occasionally, the stroma can be highly fibrous (sclerotic type) or have myxoid changes (myxoid type). The characteristics of neoplastic cell nuclei (Figure 12H,I) require particular attention, since they have been poorly analyzed. Ultrastructurally, many nuclei are elongated, with thin strips of heterochromatin attached to the nuclear membrane, moderate size nucleoli, and folds and lobules joined by fine bridges of varying length formed by the nuclear membranes and heterochromatin strips (Figure 12I), acquiring a map-like, convoluted appearance (Figure 12I). Light microscopy and conventional techniques show that the same nucleus appears as two or more irregularly aligned nuclei mistakenly believed to be several cells, except when observed at very high magnification (generally not used in routine diagnosis of this process) (Figure 12H). We have identified similar nuclear characteristics to those described for dermatofibrosarcoma in stromal neoplastic cells of cystosarcoma phyllodes of the breast.

#### 4.1.2. CD34+ Stromal Cells in Sclerotic Fibroma (Circumscribed Storiform Collagenoma)

This unencapsulated and well-circumscribed nodular dermal process, composed of broad collagen bundles with a whorled (storiform) arrangement [67] shows CD34+ stromal cells around collagen bundles [68]. Microscopically, the lesion presents an arabesque image when immunostained for CD34 and observed at low magnification (Figure 13A). Occasionally, structures resembling sensory corpuscles are seen (Figure 13B). In our observations, the somatic region of the CD34+ stromal cells is increased and the processes surrounding the collagen fibers are very long. A fact not previously considered is the presence of different size vessels, many of which are tiny and located at the center of each whorled arrangement of CD34+ stromal cells (tumor formed by perivascular CD34+ stromal cells). Thus, the vessels present virtual or evident lumen (Figure 13C) and occasional sprouting endothelial cells (Figure 13D), pass from one whorled structure to another (Figure 13E) and show some perivascular multinucleated CD34+ stromal cells (Figure 13F). Using double immunofluorescence labelling for CD34 and αSMA or Collagen I, we observed CD34+ stromal cells around vessel mural cells (Figure 13G) and Collagen I (Figure 13H), respectively.

#### 4.1.3. CD34+ Stromal Cells in Solitary Fibrous Tumor

In solitary fibrous tumors, with hyper and hypo cellular areas, spindled or round stromal cells express CD34 [69] (Figure 14A,B). These cells are arranged in multiple patterns, such as fascicular and storiform. When CD34+ stromal cells acquire a fascicular pattern, they show long, bipolar processes (Figure 14B).

### 4.2. Tumor/Tumor-Like Conditions of the Skin in Which CD34+ Stromal Cells Are a Neoplastic Component with Varying Presentation

The fibroepithelial polyp and superficial fibromyxosarcoma will be studied as examples of this section.

#### 4.2.1. CD34+ Stromal Cells in Fibroepithelial Polyp (Acrochordon)

The spindled, pleomorphic and occasional multinucleated stromal cells in the fibrovascular connective tissue that forms the core of fibroepithelial polyps show CD34 expression (Figure 14C–E) [70,71] and in some cases (above all with components resembling pleomorphic fibroma) αSMA expression.

#### 4.2.2. CD34+ Stromal Cells in Superficial Myxofibrosarcoma

Hyperchromatic, atypical, spindled cells, some with a bizarre appearance, prominent nucleoli and mitotic Figure, show a varying expression of CD34 [72]. In our observations, these cells have numerous processes (Figure 14F), present mitosis (Figure 14G) and can lose CD34 expression in some cells (Figure 14G) or throughout the tumor (Figure 14H).

### 4.3. Reactive CD34+ Stromal Cells in Tumors and Tumor-Like Conditions of the Skin Formed by Other Cell Lines

In this section we will present tumors originating in vascular wall cells, epithelial cells, Merkel cells, melanocytes and Schwann cells.

#### 4.3.1. Myopericytoma

We will first consider myopericytoma as an example of tumors with vascular neoplastic αSMA+ cells. In this tumor, the border/capsule can also be a specific example of repair, which we have studied previously [33,73]. Thus, the main purpose is to compare the expression of CD34 and αSMA in the stromal cells of the myopericytoma border/capsule with that in tumors formed by neoplastic CD34+ cells. Interestingly, the borders of tumors composed of neoplastic CD34+ cells present abundant αSMA+ cells, as occurs in solitary fibrous tumors (Figure 14I). Conversely, in benign pericytic (perivascular) tumors composed of neoplastic cells showing immunohistochemical positivity for αSMA (e.g., myopericytoma), the stromal cells in the capsule and surrounding tissues are positive for CD34 (Figure 14J). Therefore, the border of each group of tumors comprises stromal cells with an opposite expression to that of the tumor: αSMA+ stromal cells for CD34+ tumors and CD34+ stromal cells for αSMA tumors.

#### 4.3.2. CD34+SCs/TCs in the Stroma of Tumor/Tumor-Like Conditions Originating from Epithelial Cells

There are previous studies on telocytes in the stroma of basal and squamous cell carcinoma [50]. One ultrastructural study of telocytes in these tumors described very few heterocellular junctions in telocytes, which supports a possible involvement in the induction of cell-cell communication. Next, we will consider CD34+SC/TC behavior in trichoepithelioma and in the nevus sebaceous of Jadassohn as examples of this group of lesions.

##### CD34+SCs/TCs in Trichoepithelioma

Several authors have noted the presence of stromal cells expressing CD34 in tumors deriving from hair follicles, mainly to differentiate them from basal cell carcinoma [74,75,76]. These studies include trichoblastoma, trichofolliculoma, trichoadenoma and trichoepithelioma [75]. The main objective was to assess whether the CD34 staining pattern was useful for the differential diagnosis between basal cell carcinoma and trichoepithelioma. Initial studies were very promising for distinguishing both entities, since CD34 was negative in basal cell carcinoma [74] or stronger in trichoepithelioma [75]. However, other extensive observations have shown CD34 expression in stromal cells in trichoepithelioma (76%) and basal cell carcinoma (46%) [76]. In trichoblastoma, trichofolliculoma, trichoadenoma and trichoepithelioma, stromal cells expressing CD34 were observed adjacent to the external surface of tumoral islands [75]. To establish the characteristics of CD34+SCs/TCs in the stroma of these tumors, we will use trichoepithelioma as an example.

Trichoepithelioma is made up of branched tracts (cords) of basaloid cells (frond-like appearance), papillary mesenchymal bodies and horny cysts. The stroma generally presents moderate myxoid-like changes, few inflammatory cells and varying numbers of CD34+SCs/TCs arranged around the epithelial tracts (Figure 15A). These cells have elongated or ovoid nuclei in a small somatic region, from which several processes (telopodes) emerge and surround the branched epithelial tracts (Figure 15A). CD34+SCs/TCs can be fully attached to the external surface of the epithelium (Figure 15B–E) or present their somatic region in the interstitium and processes that extend to the epithelial tracts (Figure 15F,G). CD34+SCs/TCs are usually bipolar when attached to the epithelium (Figure 15B–D). Those with the somatic region in the interstitium are stellate (multipolar) (Figure 15F,G) and some have a pyriform appearance (Figure 15G). Collagen I can be observed between these cells (Figure 15H).

##### CD34+SCs/TCs in the Nevus Sebaceous of Jadassohn

Numerous CD34+SCs/TCs are observed around the sebaceous glands (Figure 16A,B), sweat glands (Figure 16C) and scant hair follicles (Figure 16D), in the nevus sebaceous of Jadassohn, which shows papillomatosis, hyperplasia of sebaceous glands located superficially and with direct openings through the epidermis (Figure 16A), eccrine and apocrine glands and few or absent hair follicles. Perivascular or interstitial CD34+SCs/TCs are not detected in the areas of papillomatosis (Figure 16E). We have also observed CD34+ cells around basaloid nests (Figure 16F) and in the dermal papilla of occasional hair follicles.

##### CD34+SCs/TCs in Seborrheic Keratosis

In seborrheic keratosis, CD34+ stromal cells are located at the bottom of the lesion (Figure 16G), but are not observed in the papillary areas with papillomatosis (Figure 16G).

#### 4.3.3. CD34+SCs/TCs in Tumors Originated from Merkel cells (Merkel Cell Carcinoma)

This primary neuroendocrine carcinoma in the skin is formed by nests, trabeculae and sheets of cells with scant cytoplasm, oval nuclei, numerous mitotic Figure and positivity (frequently perinuclear and with a dot-like appearance) for cytokeratins 20 and AE1/AE3, and neuroendocrine markers [77]. It also shows prominent peritumoral stromal cells, which express αSMA (Figure 16H). Activated CD34+SFCs/TCs are observed a short distance from the tumor components and show an increased somatic region with voluminous nuclei (Figure 16I).

#### 4.3.4. CD34+SCs/TCs in Tumors Originated from Melanocytes and Schwann Cells

Our observations on CD34+SCs/TCs in melanocytic nevi and malignant melanomas coincide with previous studies on the presence of CD34+ connective cells in melanocytic nevi and absence in melanomas [78].

In compound and dermal melanocytic nevi, nevi nests or isolated nevic cells are surrounded by CD34+SCs/TCs (Figure 17A–C), which are scarce in the most superficial areas of the lesion and very numerous in deeper areas (in the reticular dermis). Thus, in some regions, CD34+SCs/TCs can be more abundant than nevic cells (Figure 17A). Generally, CD34+SCs/TCs show bipolar morphology with long, thin telopodes (Figure 17C). Pigmented particle storage is observed in some of these cells [5]. CD34+SCs/TCs are also present in common and cellular blue nevi. In these cases, the bipolar CD34+SCs/TCs can contain a greater amount of melanin pigment. CD34+SC/TC processes in the common type are frequently parallel to elongated dermal melanocytes and the surface of the epidermis. In cellular blue nevus, the connective tissue between the compact, irregular tumor nests shows CD34+ cells, whose processes penetrate the peripheral areas of the tumor nests, dissecting small groups of tumor cells.

We have previously studied CD34+SCs/TCs in tumors originating from Schwann cells (neurofibroma and granular cell tumor) [19]. We confirm that the number of CD34+SCs/TCs can be higher than the number of Schwann cells in neurofibroma and that CD34+SCs/TCs surround granular cells in the granular cell tumor. Now, we present their characteristics in immunofluorescence (neurofibroma, Figure 17D) and in electron microscopy (granular cell tumor, Figure 17E).

## 5. General Considerations and Required Future Studies about CD34+SCs/TCs in Normal and Pathological Skin

In our review of the characteristics, distribution and behavior of CD34+SCs/TCs in normal and pathological skin, to which we have added our own observations, the following facts are remarkable. In normal skin, we confirm previous studies by other authors on the characteristics and distribution of CD34+SCs in skin appendages and other dermal components [13,35,36,37,38,39,40,41,42,43,44]. In addition, we have specified and/or contributed the following: (a) the relationship of CD34+SCs/TCs with blood and lymphatic vessels in the two horizontal vascular plexuses and connecting vessels in the dermis, (b) the absence of CD34+SCs/TCs around vessels and in the interstitium of the papillary dermis, which may contribute to preventing excessive repair phenomena in simple erosions of the skin, as occurs in superficial areas of the mucosa of the intestine and gallbladder, and affect the formation of tumor stroma [54] and the behavior of melanomas, depending on the level of invasion (e.g., whether or not the papillary dermis is exceeded) and c) the presence around the bulge region of the hair follicle of isolated groups of very small CD34+ stromal cells, which may correspond to the mesenchymal stromal/stem cells isolated from the hair follicle dermal sheath [55]. Future studies are required in the last two sections.

We have reviewed the behavior of CD34+SCs/TCs in one or several diseases in each of the principal non-tumoral and tumoral histopathological patterns of the skin. The purpose was to obtain a comprehensive outline, without making the work overly long. Although there are important studies on the behavior of CD34+SCs/TCs in non-tumoral pathological processes of the skin, such as systemic fibrosis [38,47,48,56,57,58,59,60,61] and psoriasis [49], this is not the case for most non-tumoral pathological processes.

The participation of CD34+SCs/TCs in the organization of the extracellular matrix explains the important role of these cells in fibrosing/sclerosing diseases—as the Italian school has demonstrated in systemic sclerosis (scleroderma) [38,47,48,56,57,58,59,60,61]—as well as in the basophilic degeneration of collagen and mucinosis. In scleroderma, we show the coexpression of CD34 and αSMA in some stromal cells, which suggests the participation of CD34+SCs/TCs as a source of myofibroblasts in this lesion. Likewise, we highlight the association of CD34+SCs/TCs with degenerative fibers in the basophilic degeneration of collagen, and the presence of spindle-shaped, stellate and bulky vacuolated CD34+ stromal cells in cutaneous myxoid cysts, as an example of local mucinosis.

In addition to psoriasis, in which CD34+Sc/TCs have been studied by other authors [36], we have contributed the behavior of reactive CD34+SCs/TCs in examples of several histopathological patterns of non-tumoral processes of the skin. The examples include erythema multiforme, pemphigus, Hailey–Hailey disease, lichen planus, bullous pemphigoid, granuloma annulare, leukocytoclastic and lymphocytic vasculitis, folliculitis/perifolliculitis, rosacea and verruca vulgaris. In general, CD34+SCs/TCs surround perivascular inflammatory infiltrates in the reticular dermis but are absent when these infiltrates are located in the papillary dermis—a reflection of the distribution of these cells in normal conditions. Perivascular and interstitial CD34+SCs/TCs are usually absent in large inflammatory infiltrates and in granulomas but present around them. Several factors are known to be involved in immune cell control in the interstitial migration of CD4+ T lymphocytes in the inflamed dermis [79,80,81] and CD34+SCs/TCs in the reduction of the inflammatory response in skin wound healing models [45]. Future studies are required to typify inflammatory cells arranged around vessels and surrounded by perivascular CD34+SCs/TCs in the reticular dermis affected by pathological processes, and to evaluate the possible role of CD34+SCs/TCs in retaining and modulating inflammatory cells in the perivascular spaces.

The expression of CD34 has been explored more in stromal cells of tumors and tumor-like conditions than in the entities outlined above, primarily for diagnostic purposes (47–62). In the examples with neoplastic CD34+ stromal cells, we paid particular attention to the characteristics of their nuclei in dermatofibrosarcoma—barely considered in the literature—and the CD34 arrangement around vessels in sclerosing fibroma (perivascular CD34+ stromal cell collagenoma). In examples of cases in which stromal cells are the neoplastic component whose CD34 expression varies, further studies are required for the loss of CD34 expression, degenerative phenomena or both possibilities in the stromal cells of myxofibrosarcoma.

In examples of reactive CD34+SCs/TCs in skin tumors and tumor-like conditions formed by cell lines other than CD34+ stromal cells, we highlight (a) findings supporting the hypothesis that in tumors formed by αSMA+ cells (e.g., pericytic/myopericytic tumors and leiomyomas) or by CD34+ cells (e.g., solitary fibrous tumor), reactive stromal cells in the borders express the other marker, (b) the epithelial tumors can present CD34+ or αSMA+ stromal cells, or even both types of cells, depending on the tumoral region examined, (c) specific distribution of CD34+SCs/TCs in some lesions, as occurs in nevus sebaceous of Jadassohn and seborrheic keratosis, (d) CD34+SC/TC changes in size and morphology in certain tumors, such as trichoepithelioma and Merkel cell tumor, and (e) an important increase in CD34+SC/TC numbers in some cases of melanocytic nevi, neurofibromas and granular cell tumors.

A remarkable finding from our observations on CD34+SCs/TCs in the skin is the partial or total absence of CD34+ stromal cells in some pathological processes, which may be due to (a) the existence of regions without CD34+SCs/TCs in normal conditions, as occurs in the papillary dermis, (b) loss of CD34 expression in CD34+SCs/TCs and gain of other markers, including αSMA expression, which is the case of some stromal cells in systemic sclerosis and in the stroma of some tumors, and (c) degenerative phenomena, which occurs in some stromal cells in systemic sclerosis and in neoplastic cells in myxofibrosarcoma. Further studies are required in these issues, mainly in the gain of other markers by CD34+SCs/TCs during the development of pathological processes of the skin. Future studies should also consider the practical applicability of the findings described for CD34+SCs/TCs in the normal and pathologic skin, including their possible therapeutic implications.

In conclusion, we have reviewed the current state of knowledge about CD34+SCs/TCs in stromal and pathological skin, including our own observations. While CD34+SCs/TCs have been widely explored in normal conditions, they have only been studied in a few pathological processes of the skin. We have contributed examples of entities included in the principal non-tumoral and tumoral histopathological patterns of the skin to obtain a broad overview of the behavior of CD34+SCs/TCs in these conditions and to lay the groundwork for future studies, including the practical applicability.

## Figures and Tables

**Figure 1 ijms-22-07342-f001:**
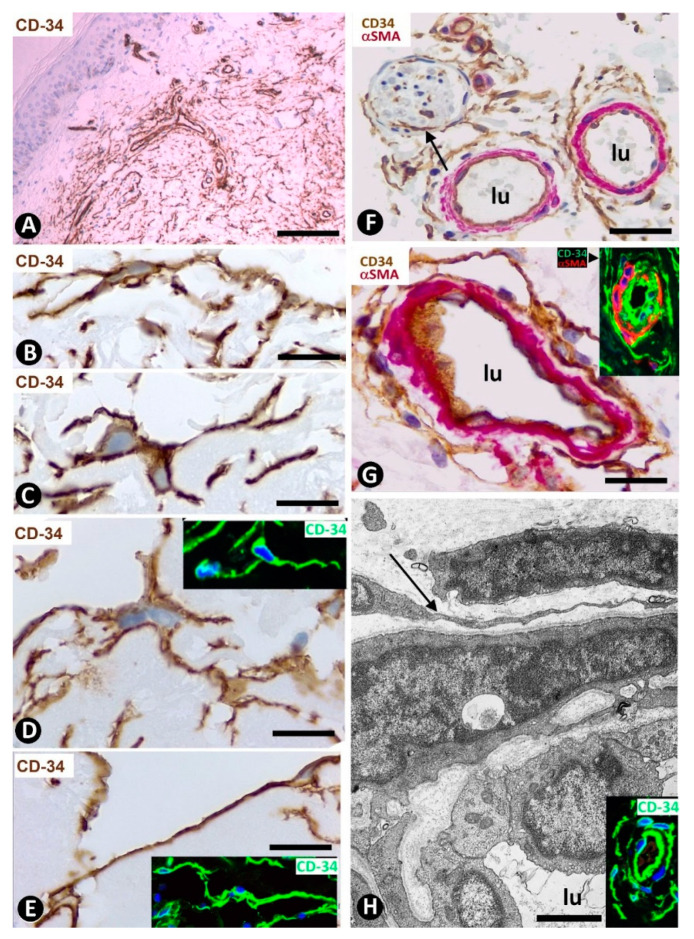
CD34+SCs/TCs in normal skin. (**A**–**E**) CD34 immunochemistry. Hematoxylin counterstain. (**F**,**G**) Double immunochemistry for CD34 (brown) and αSMA (red). (**H**) Ultrathin section. Uranyl acetate and lead citrate, Inserts of (**D**,**E**,**H**): Immunofluorescence labelling for CD34 (green). Insert of (**G**) Double immunofluorescence labelling for CD34 (green) and αSMA (red). DAPI counterstain. A: Panoramic view, in which a greater number of CD34+SCs/TCs is observed in the reticular dermis. (**B**–**E**) Morphologic characteristics of dermal CD34+SCs/TCs. Note a small somatic region from which long, thin bipolar or multipolar processes (telopodes) emerge. (**F**,**G**) Presence of CD34+SCs/TCs (brown) around vessels. The vascular mural cells are stained red. In (**F**), CD34+SCs/TCs are observed surrounding a nerve (arrow). (**H**) Ultrastructural image of a telocyte (arrow) around a vessel. Inserts show similar images in immunofluorescence to the corresponding Figure in immunochemistry. Vessel lumen:lu. Bar: (**A**) 150 µm, (**B**–**E**,**G**) 25 µm, (**F**) 80 µm, (**H**) 3 µm.

**Figure 2 ijms-22-07342-f002:**
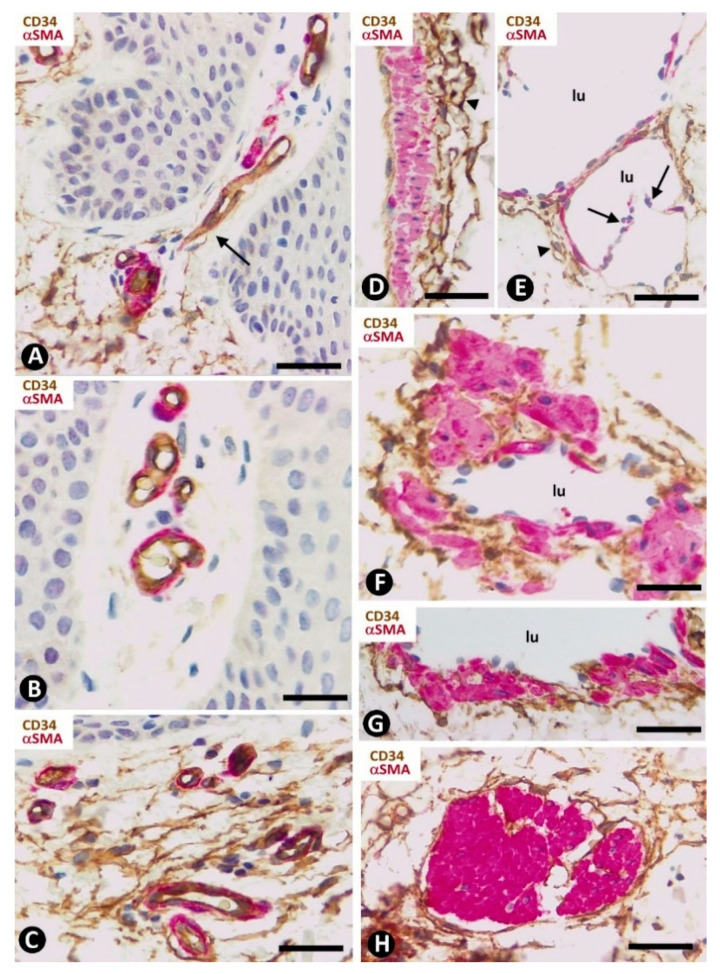
CD34+SCs/TCs in normal skin. Double immunochemistry for CD34 (brown) and αSMA (red). (**A**,**B**) Absence of interstitial and perivascular CD34+SCs/TCs in the papillary dermis. Note a vessel in (**A**), in which the perivascular CD34+SCs/TCs are absent as the vessel enters the papillary dermis (arrow). (**C**) Vessels in the upper horizontal plexus and numerous CD34+SCs/TCs in perivascular and interstitial location. (**D**,**E**) The wall of an artery (**D**) and veins (**E**) in the plexus located in the dermal subcutaneous junction. Several layers of CD34+SCs/TCs are observed in the arterial adventitia (**D**, arrowhead) and in smaller numbers around veins (**E**, arrowhead). Note the presence of cusped valves in the latter (arrows). (**F**,**G**) CD34+SCs/TCs (brown) surrounding groups of smooth muscle cells (red) in pre-collector lymphatic vessels, in which the endothelial cells do not express CD34. **H**: CD34+SCs/TCs around fascicles of arrector pili muscle. Vein (**E**) and lymphatic (**F**,**G**) lumen: lu. Bar: (**A**,**C**,**F**,**G**,**H**) 45 µm, (**B**) 55 µm, (**D**,**E**) 80 µm.

**Figure 3 ijms-22-07342-f003:**
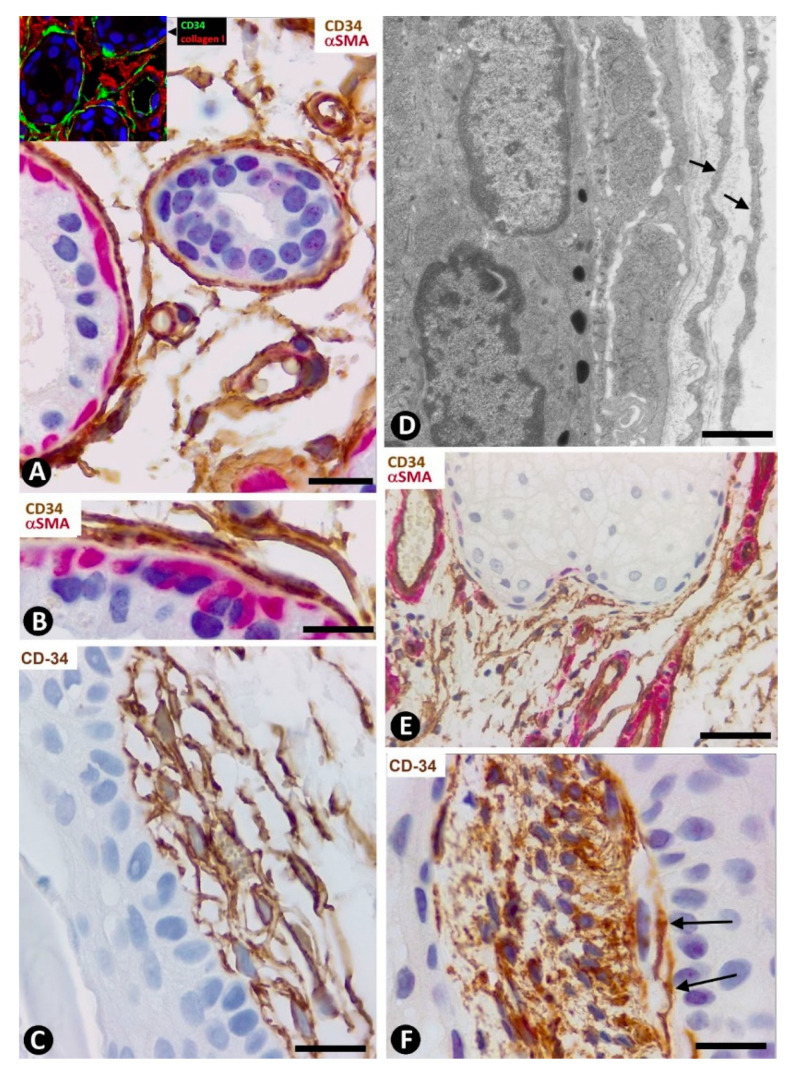
CD34+SCs/TCs around sweat glands, hair follicles and sebaceous glands. (**A**,**B**,**E**) Double immunochemistry for CD34 (brown) and αSMA (red). Hematoxylin counterstain. (**C**,**F**) Immunochemistry for CD34. Hematoxylin counterstain. (**D**) Ultrathin section. Uranyl acetate and lead citrate. (**A**,**B**) CD34+SCs/TCs (brown) are observed around the sweat glands at different magnifications. Myoepithelial cells are stained red. (**C**) Several layers of CD34+SCs (brown) around a hair follicle. (**D**) Ultrastructural image of telopodes of two telocytes (arrows) surrounding a hair follicle. (**E**) CD34+SCs/TCs (brown) around a sebaceous gland. (**F**) A cluster of very small, densely grouped CD34+ stromal cells with multiple intricate processes between the bulge region of a hair follicle and a sebaceous gland. Note CD34+SCs/TCs (arrows) interposed between the cluster of small CD34+ stromal cells and the hair epithelium. Bar; (**A**–**C**,**F**) 25 µm, (**D**) 3 µm, (**E**) 80 µm.

**Figure 4 ijms-22-07342-f004:**
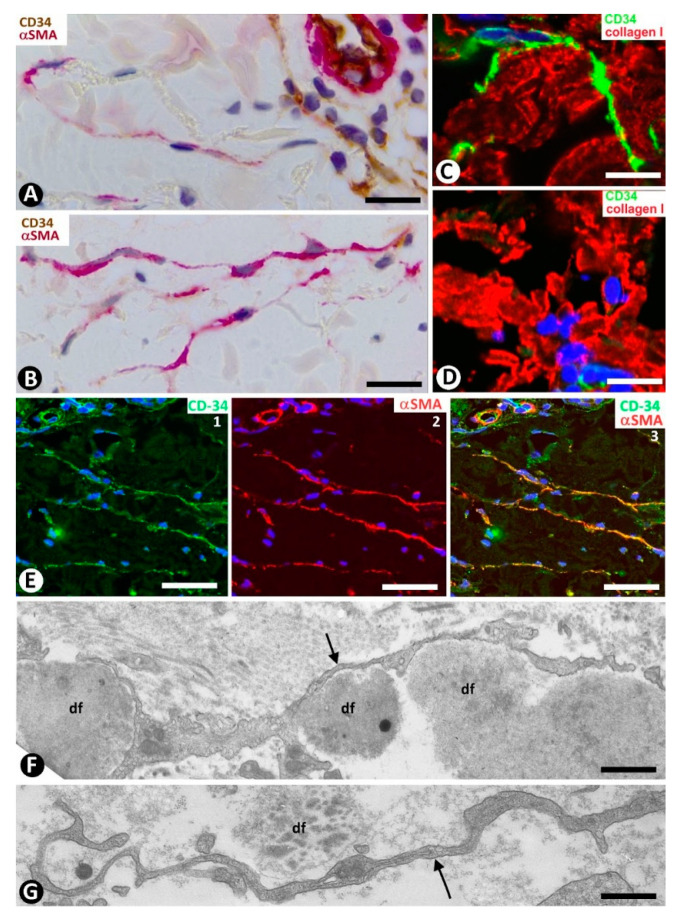
Scleroderma (**A**–**E**) and basophilic degeneration of the collagen (**F**,**G**). (**A**,**B**) Double immunochemistry for CD34 (brown) and αSMA (red). Hematoxylin counterstain. (**C**,**D**) Double immunofluorescence labelling for CD34 (green) and Collagen I (red). DAPI counterstain. (**E**) (1,2,3): Confocal microscopy, frontal view. Immunofluorescence labelling for CD34 (green) and αSMA (red). (**F**,**G**) Ultrathin sections. Uranyl acetate and lead citrate. (**A**–**E**) Stromal cells expressing CD34 (brown in (**A**) and green in **C**,**E**) or αSMA (red) in scleroderma. Note the expression of CD34 around Collagen I (red) in C and the regional absence of CD34 expression in (**D**). Stromal cells co-expressing CD34 (green) and αSMA (red) are observed in (**E**) (1,2,3). (**F**,**G**) Ultrastructural images of telopodes (arrows) closely associated with several degenerative dermal fibers (df) in basophilic degeneration of the collagen. Bar: (**A**,**B**) 25 µm, (**C**–**G**) 2.5 µm.

**Figure 5 ijms-22-07342-f005:**
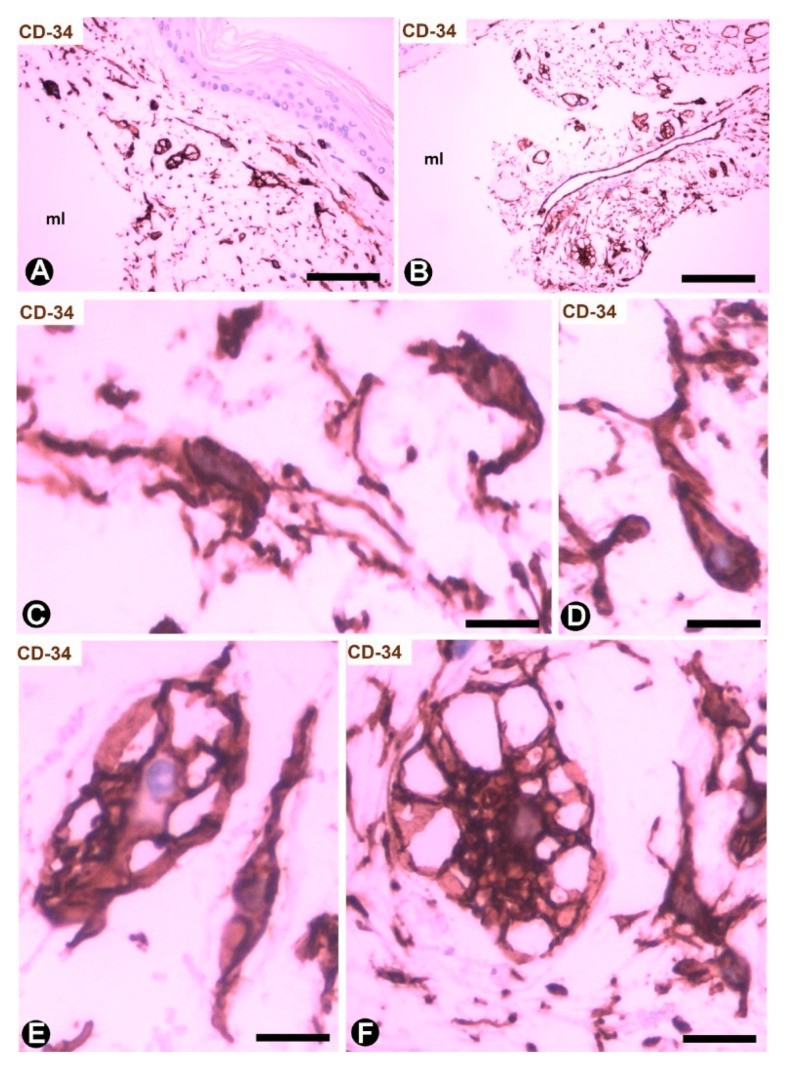
CD34+SCs/TCs in cutaneous myxoid cyst. CD34 immunochemistry; hematoxylin counterstain. (**A**,**B**) TCs/CD34+SCs are present in loose connective tissue of the dermis with myxoid lagoons (mL). (**C**,**D**) A spindle-shaped, stellate, pyriform or irregular morphology is observed in many of these cells. (**E**,**F**) Presence of bulky and multi-vacuolated CD34+ mononuclear cells in the myxoid areas. Bar: (**A**,**B**) 80 µm, (**C**–**F**) 15 µm.

**Figure 6 ijms-22-07342-f006:**
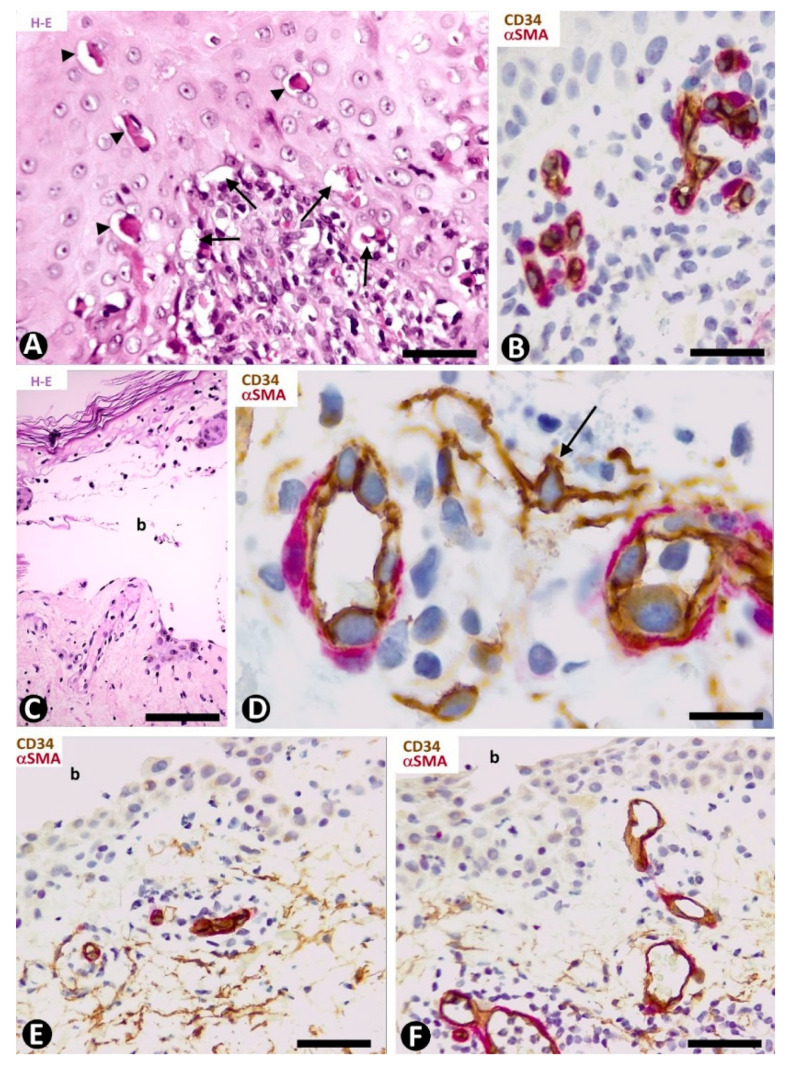
CD34+SCs/TCs in Erythema multiforme (**A**–**D**) and Pemphigus (**E**,**F**). (**A**,**C**) Hematoxylin-eosin staining. (**B**,**D**–**F**) Double immunochemistry for CD34 (brown) and αSMA (red). Hematoxylin counterstain. (**A**,**B**) Early lesion of erythema multiforme, with hydropic degeneration of basal cells (arrows), cytoid bodies (arrowheads) and lymphocytic infiltrate (**A**), in which CD34+SCs/TCs are not observed in the superficial areas (**B**). (**C**,**D**) A blister (**C**, b) in erythema multiforme with the presence of CD34+SCs/TCs (arrow) in the underlying dermis (**D**). (**E**,**F**) Intraepidermal blister (b) in which the roof is not shown. In the underlying dermis CD34+SCs/TCs are present (**E**) or absent (**F**) around some vessels. Bar: (**A**) 55 µm, (**B**,**E**,**F**) 45 µm, (**C**) 80 µm, (**D**) 25 µm.

**Figure 7 ijms-22-07342-f007:**
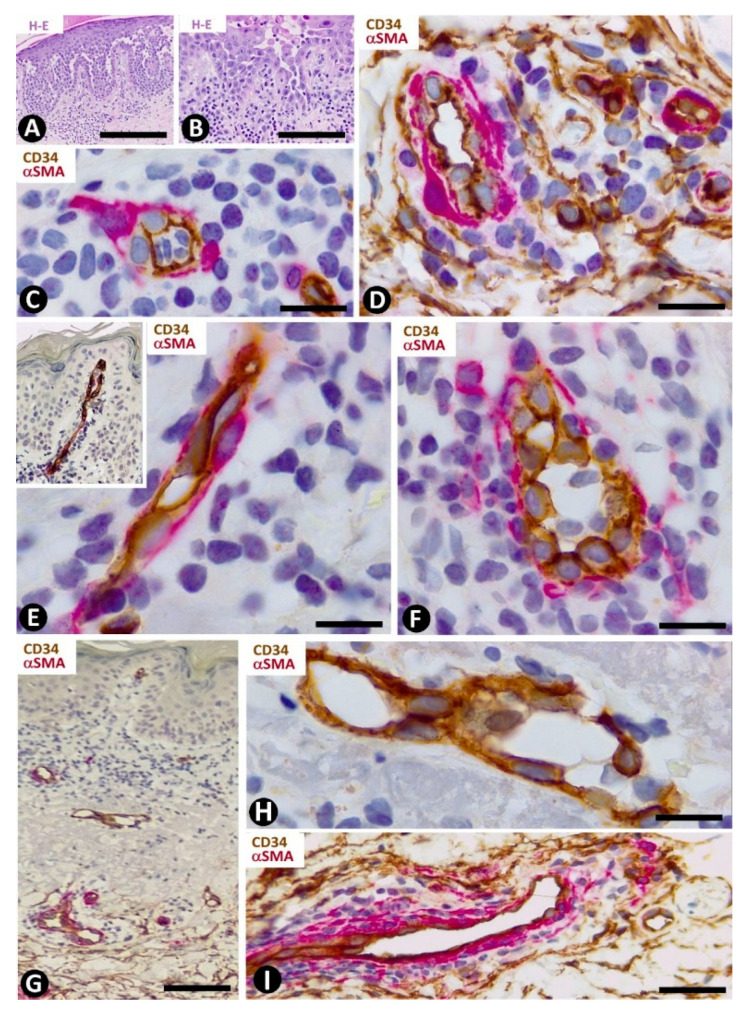
CD34+SCs/TCs in Hailey–Hailey disease (**A**–**D**) and lichen planus (**E**–**I**). (**A**,**B**) Hematoxylin and eosin staining. (**C**–**I**) Double immunochemistry for CD34 (brown) and αSMA (red). Hematoxylin counterstain. (**A**–**D**) Hailey–Hailey disease, with acantholysis (**A**,**B**), in which perivascular CD34+SCs/TCs are absent in the subepidermal inflammatory infiltrate (**C**) and present in underlying areas, surrounding mononuclear cells (**D**). (**E**–**I**) Lichen planus. Perivascular and interstitial CD34+SCs/TCs are absent in superficial (**E**–**G**) and intermediate (**G**,**H**) dermal regions, and present in deeper ones (**I**). In the latter, note perivascular CD34+SCs/TCs (brown) around inflammatory cells. Bar: (**A**,**B**,**G**) 150 µm, (**C**,**D**,**E**,**F**,**H**) 25 µm, (**I**) 80 µm.

**Figure 8 ijms-22-07342-f008:**
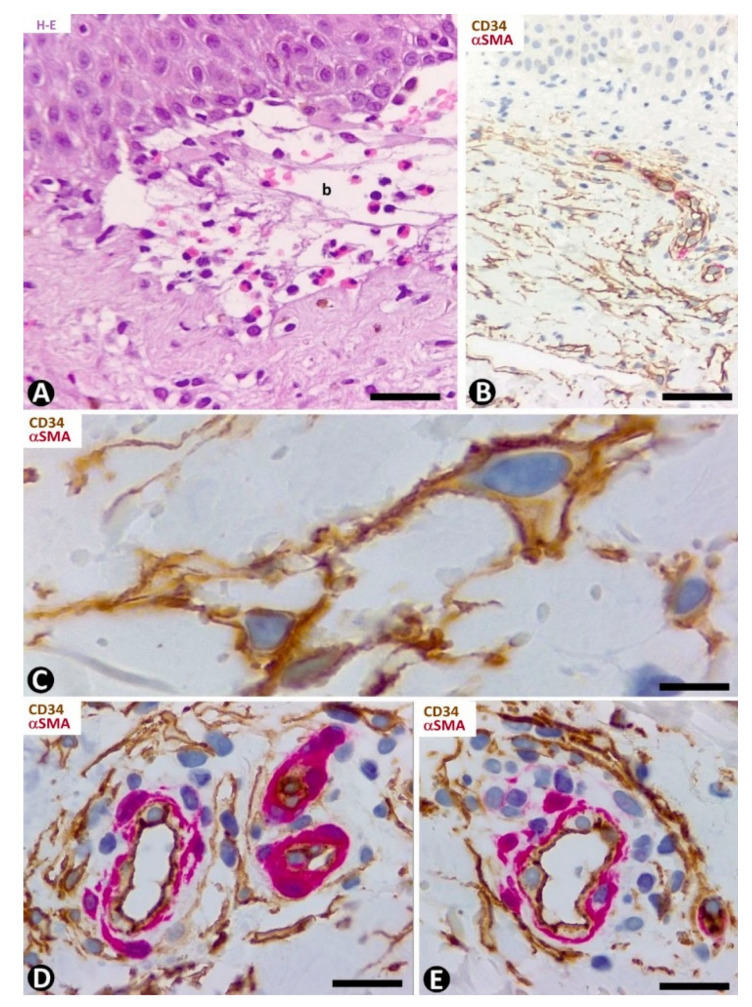
CD34+SCs/TCs in bullous pemphigoid. (**A**) Hematoxylin–eosin staining. (**B**–**E**) Double immunochemistry for CD34 (brown) and αSMA (red). Hematoxylin counterstain. (**A**) Subepidermal blister (b), with the presence of eosinophils in the blister cavity. (**B**) CD34+SCs/TCs (brown) in the superficial dermal layer. (**C**) CD34+SCs/TCs observed at high magnification in the interstitium of the superficial dermal layer. (**D**,**E**) Perivascular CD34+SCs/TCs surrounding a slight lymphohistiocytic infiltrate. Bar: (**A**) 55 µm, (**B**) 80 µm, (**C**) 15 µm, (**D**,**E**) 25 µm.

**Figure 9 ijms-22-07342-f009:**
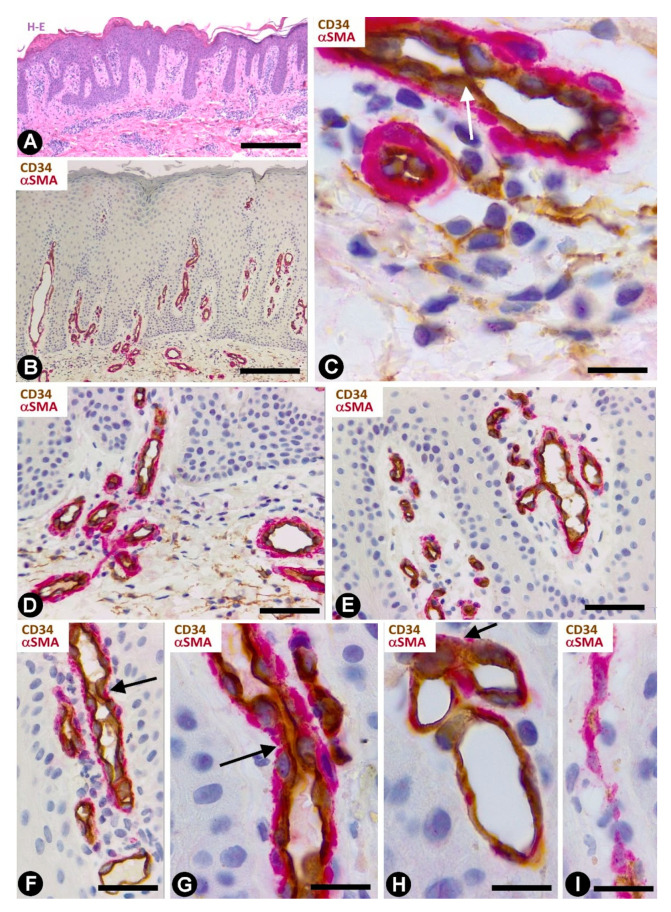
CD34+SCs/TCs in psoriasis. (**A**) Tissue section stained with hematoxylin and eosin, showing elongated rete ridges and a superficial perivascular inflammatory lymphocytic infiltrate. (**B**–**I**) Double immunochemistry for CD34 (brown) and αSMA (red). Hematoxylin counterstain. Vessels in the superficial layer extend and penetrate the papillary dermis between the elongated rete ridges (**B**, arrows). Some CD34+SCs/TCs are observed around vessels between the superficial inflammatory infiltrates (**C**), but are absent in the papillary dermis (**D**–**I**). Vessels in the papillary and superficial dermis present interendothelial apical (**C**,**F**, arrows) and planar (**G**, arrow) contacts, folds with incarcerated pericytes (**H**, arrow) and occasional regressive phenomena (**I**). Bar: (**A**,**B**) 150 µm, (**C**,**G**,**H**,**I**) 25 µm, (**D**,**E**,**F**) 80 µm.

**Figure 10 ijms-22-07342-f010:**
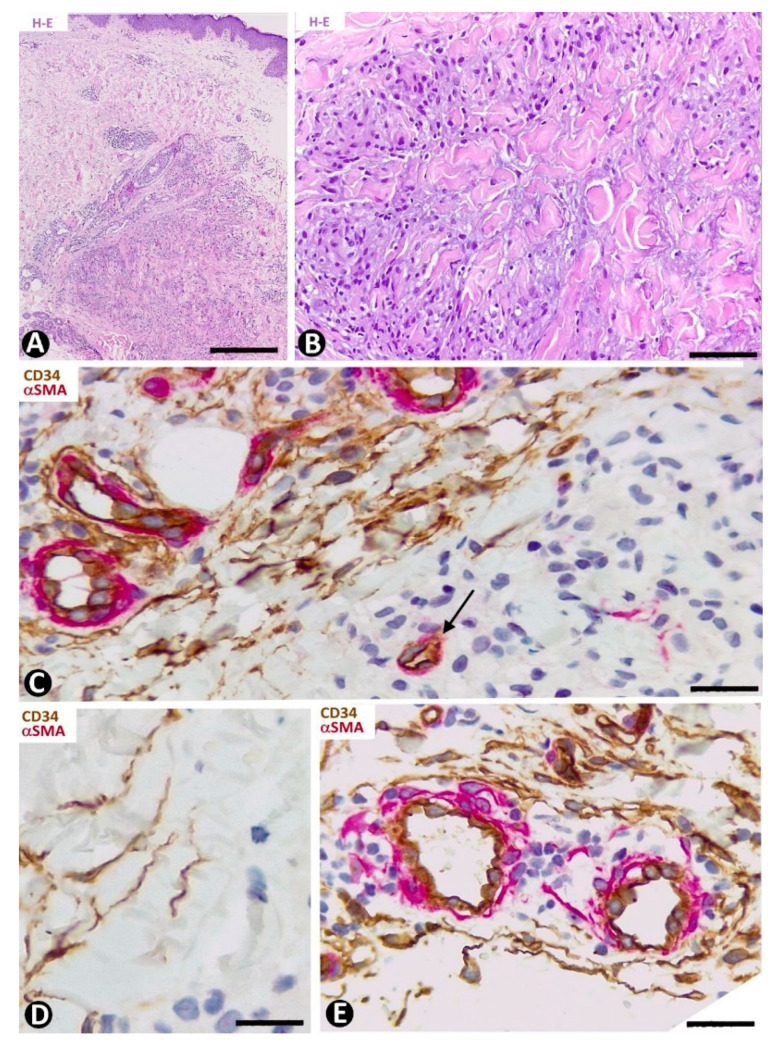
CD34+SCs/TCs in the granuloma annulare. (**A**,**B**) Hematoxylin–eosin staining. (**C**–**E**) Double immunochemistry for CD34 (brown) and αSMA (red). Hematoxylin counterstain. (**A**,**B**) Images at different magnifications, showing dermal location, palisading inflammatory infiltrate and mucin deposition. (**C**,**D**) Perivascular and interstitial CD34+SCs/TCs (brown) around the granuloma. Note in C the absence of CD34+SCs/TCs around a vessel within the granuloma (arrow). (**E**) Small accumulation of lymphocytes between perivascular CD34+SCs/TCs (brown) and vessel mural cells (red). Bar: (**A**) 150 µm, (**B**) 80 µm, (**C**,**E**) 55 µm, (**D**) 45 µm.

**Figure 11 ijms-22-07342-f011:**
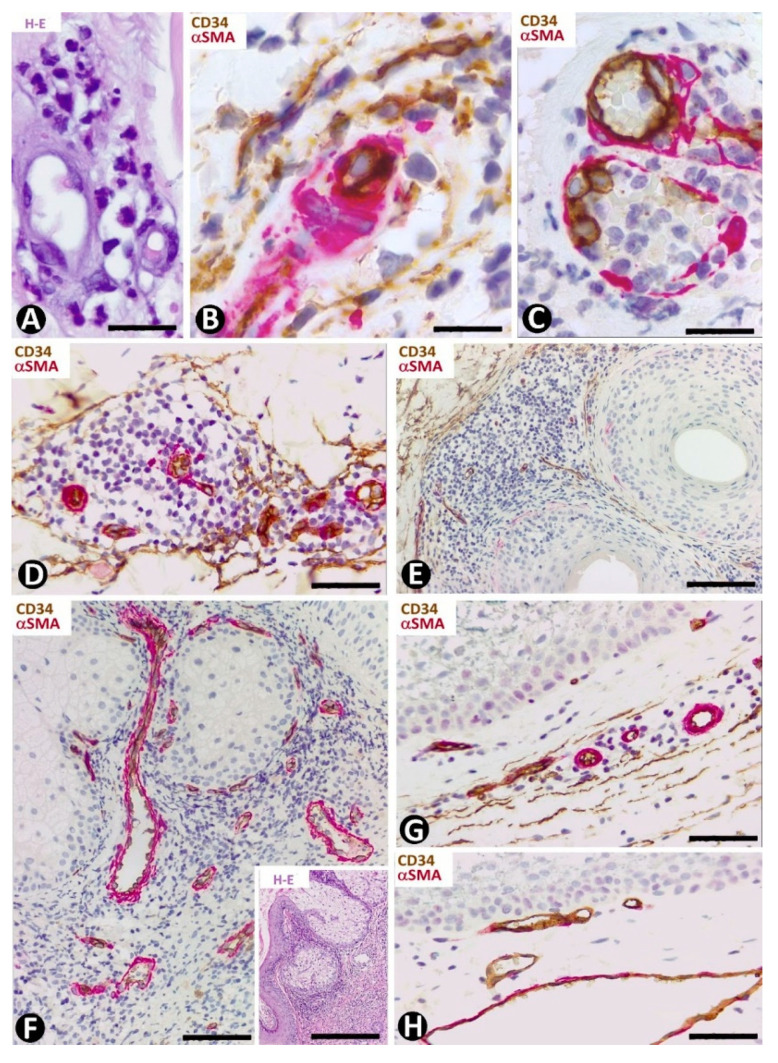
CD34+SCs/TCs in leukocytoclastic (**A**–**C**) and lymphocytic (**D**) vasculitis, perifolliculitis (**E**), rosacea (**F**) and in verruca vulgaris (**G**,**H**). (**A**) and insert of (**F**): Hematoxylin–eosin staining. (**B**–**H**) Double immunochemistry for CD34 (brown) and αSMA (red). Hematoxylin counterstain. (**A**–**C**) Leukocytoclastic vasculitis with nuclear dust (**A**) and with conservation (**B**) or loss (**C**) of CD34+SCs/TCs (brown) around vessels. Note in (**C**) numerous intravascular neutrophils in a vessel (arrow) and degenerative phenomena in endothelial cells with partial absence of them. (**D**) Lymphocytic infiltrate between vascular mural cells (red) and CD34+SCs/TCs (brown). (**E**,**F**) Inflammatory infiltrate with absence of CD34+SCs/TCs in alopecia (perifollicular) (**E**) and in rosacea (around sebaceous glands) (**F**). (**G**,**H**) Verruca vulgaris with presence (**G**) or absence (**H**) of interstitial and perivascular CD34+SCs/TCs (brown). Bar: (**A**,**B**,**C**) 45 µm, (**D**,**F**,**G**,**H**) 80 µm, (**E**) 150 µm.

**Figure 12 ijms-22-07342-f012:**
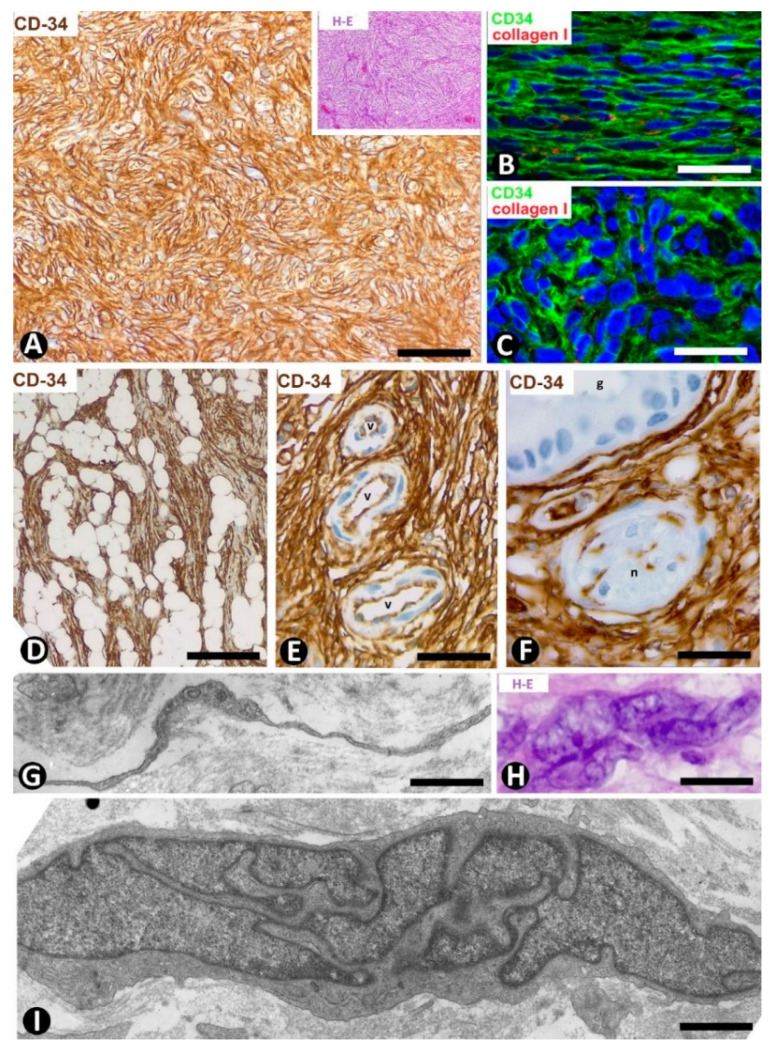
CD34+ stromal cells in dermatofibrosarcoma protuberans. (**A**,**D**,**F**) Immunochemistry for CD34. Hematoxylin counterstain. Insert of (**A**), and (**H**): Hematoxylin–eosin staining. (**B**,**C**) Double immunofluorescence labelling for CD34 (green) and Collagen I (red). DAPI counterstain. (**G**,**I**) Ultrathin sections. Uranyl acetate and lead citrate. (**A**) Proliferation of CD34+stromal cells in a storiform or intersecting (cartwheel) pattern. Insert of (**A**) Similar image in hematoxylin–eosin staining. (**B**,**C**) CD34+ neoplastic stromal cells (green) densely packed with low quantity of Collagen I (red). (**D**) Tumor extending into the subcutaneous adipose tissue. (**E**,**F**): Vessels (**E**, v) and sweat glands (**F**, g) and a nerve (**F**, n) are observed between masses of CD34+ neoplastic stromal cells. (**G**) A typical process of neoplastic cells with telopode characteristics (arrow). (**H**,**I**) Nuclei of neoplastic cells showing a characteristic convoluted and map-like appearance at high magnification in hematoxylin–eosin staining (**H**) and ultrastructurally (**I**). Bar: (**A**,**D**,**E**) 80 µm, (**B**,**C**,**F**) 45 µm, (**G**,**I**) 4 µm, (**H**) 15 µm.

**Figure 13 ijms-22-07342-f013:**
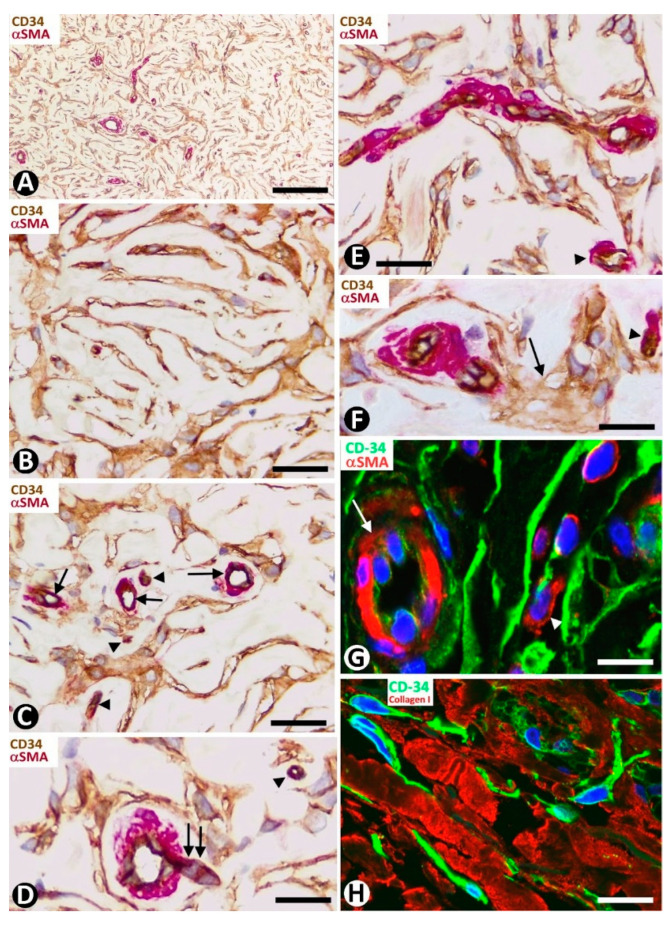
CD34+ stromal cells in sclerotic fibroma (circumscribed storiform collagenoma). (**A**–**F**) Double immunochemistry for CD34 (brown) and αSMA (red). Hematoxylin counterstain. (**G**,**H**) Double immunofluorescence labelling for CD34 (green) and αSMA (G, red) and Collagen I (**H**, red). DAPI counterstain. (**A**) Voluminous CD34+ stromal cells with a whorled (storiform) arrangement are observed around vessels. (**B**) Occasional lamellar arrangement of the CD34+ stromal cells resembling a cutaneous sensory corpuscle. (**C**–**F**) Vessels (arrows in **C**–**E** and arrowheads) at the centre of the cluster of CD34+ stromal cells that form whorls. Note that some vessels are thin, with a virtual lumen (arrowheads), present sprouting endothelial cells (**D**, double arrow), pass from one whorled structure to another (**E**) and show some perivascular multinucleated CD34+ stromal cells (**F**, arrow). (**G**,**H**) CD34+ stromal cells (green) around a vessel (mural cells: red, arrow) (**G**) and collagen I (red) (**H**). Bar: (**A**) 150 µm, (**B**–**H**) 25 µm.

**Figure 14 ijms-22-07342-f014:**
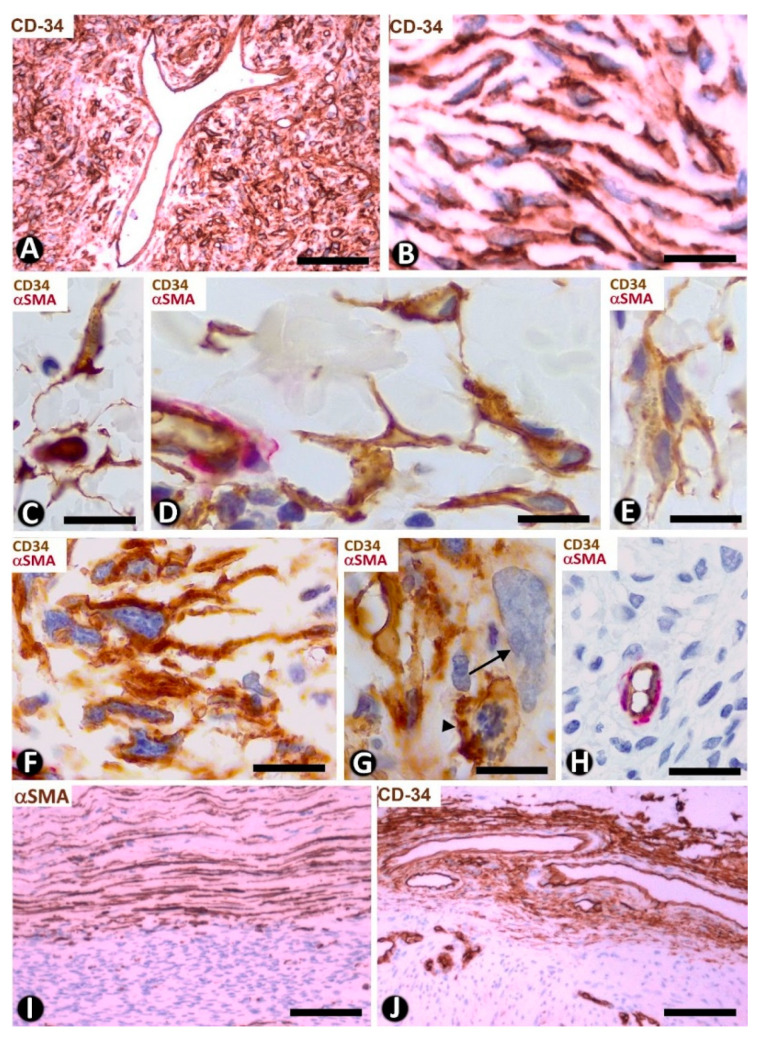
CD34+ stromal cells in a solitary fibrous tumor (**A**,**B**), fibroepithelial polyp (**C**–**E**), myxofibrosarcoma (**F**–**H**) and capsules in a solitary fibrous tumor (**I**) and in myopericytoma (**J**). (**A**,**B**,**J**) Immunochemistry for CD34 (brown). Hematoxylin counterstain. (**C**–**H**) Double immunochemistry for CD34 (brown) and αSMA (red). (**I**): Immunochemistry for αSMA (brown). Hematoxylin counterstain. (**A**,**B**): Stromal cells expressing CD34 in a solitary fibrous tumor. (**C**–**E**) Spindled and pleomorphic CD34+ stromal cells in fibroepithelial polyps. Note in (**E**) a CD34+ multinucleated stromal cell. (**F**–**H**) Stromal cells expressing CD34 or not in myxofibrosarcoma. Note in (**F**) long processes in the CD34+ stromal cells, in (**G**) a CD34+ stromal cell in mitosis (arrowhead) and another pleomorphic cell with scarce expression of CD34 (arrow), and in (**H**) absence of CD34 expression. (**I**,**J**) Capsules of a solitary fibrous tumor with stromal cells expressing αSMA (**I**) and in a myopericytoma with stromal cells expressing CD34 (**J**). Bar: (**A**,**I**,**J**) 80 µm, (**B**–**G**) 25 µm, (**H**) 45 µm.

**Figure 15 ijms-22-07342-f015:**
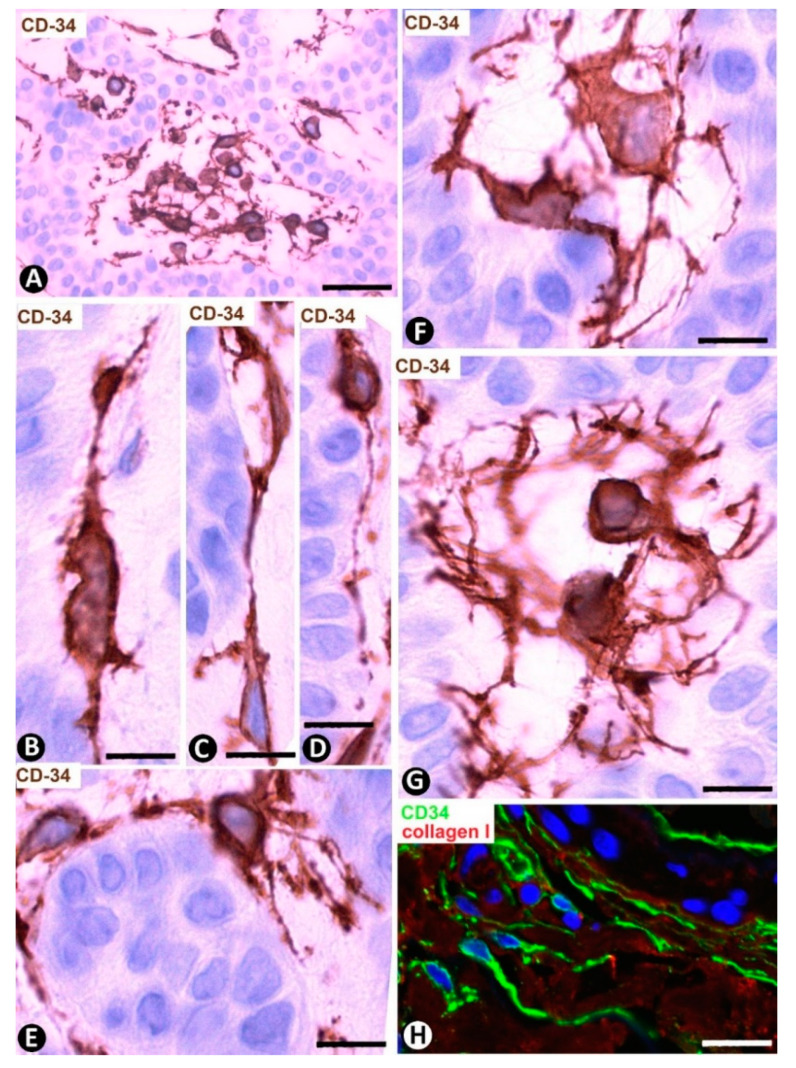
CD34+SCs/TCs in the stroma of trichoepithelioma. (**A**–**G**) CD34 immunochemistry; hematoxylin counterstain. (**H**): Double immunofluorescence labelling for CD34 (green) and Collagen (**I**) (red). (**A**) CD34+SCs/TCs are observed around the epithelial tracts. (**B**–**E**) Presence of CD34+ cells attached to the epithelium. Observe that the attached cells tend to acquire a bipolar aspect with long telopodes. (**F**,**G**) CD34+SCs/TCs with the somatic region located in the interstitium. Note that these cells are usually multipolar with telopodes that extend to the epithelial tracts. (**H**) CD34+SCs/TCs (green) are observed around Collagen I (red). Bar: (**A**) 45 µm, (**B**–**G**) 15 µm, (**C**) 15 µm, (**H**) 80 µm.

**Figure 16 ijms-22-07342-f016:**
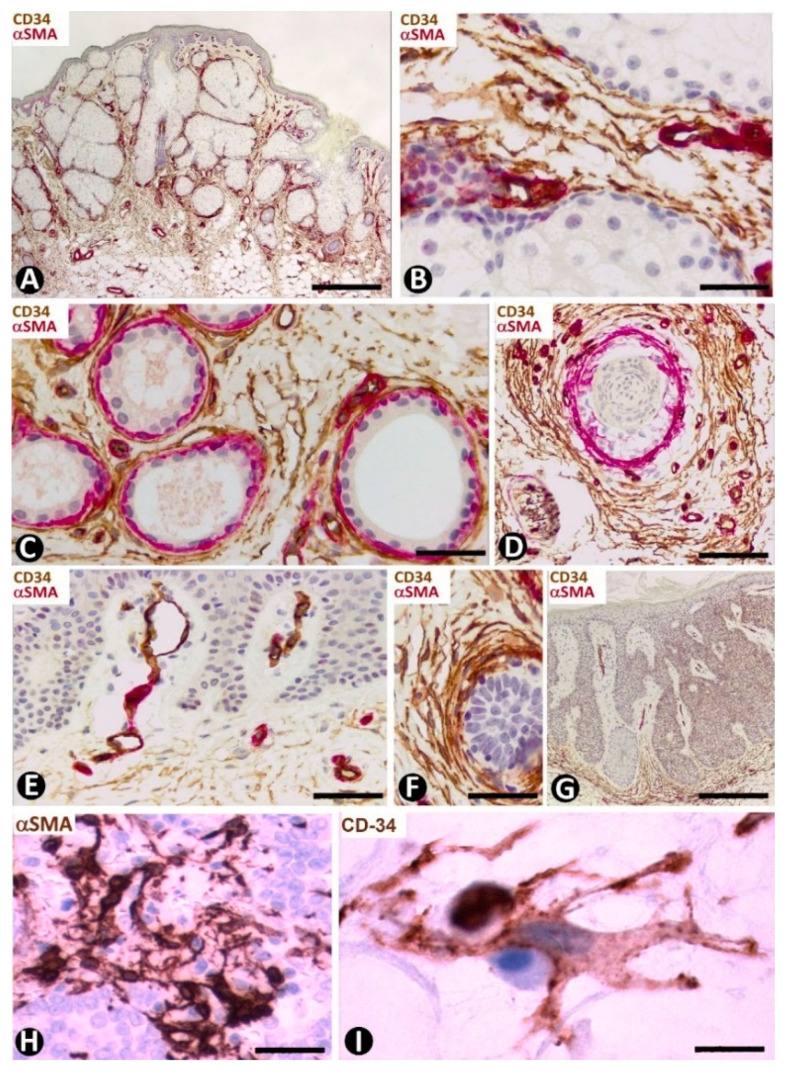
Nevus sebaceous of Jadassohn (**A**–**F**), seborrheic keratosis (**G**) and Merkel cell carcinoma (**H**,**I**). (**A**–**G**) Double immunochemistry for CD34 (brown) and αSMA (red). Haematoxylin counterstain. (**H**,**I**) Immunochemistry for αSMA (**H**) and CD34 (**I**). (**A**) Panoramic view showing hyperplasia of superficially located sebaceous glands. Note that one gland opens through the epidermis. (**B**–**D**): Numerous CD34+SCs/TCs around sebaceous glands (**B**), sweat glands (**C**) and an involutive hair follicle in the lesion (**D**). (**E**) Absence of perivascular and interstitial CD34+SCs/TCs in areas with papillomatosis. (**F**) Abundant CD34+SCs/TCs around a basaloid proliferation. (**G**) Seborrheic keratosis in which CD34+ stromal cells are located at the bottom of the lesion, but are not observed in papillary areas with papillomatosis. (**H**,**I**) Merkel cell carcinoma with stromal cells, which show an increased somatic region and voluminous nuclei, and express αSMA (**H**) and CD34 (**I**). Bar: (**A**,**G**) 150 µm, (**B**,**H**) 45 µm, (**C**–**F**) 80 µm, I 15 µm.

**Figure 17 ijms-22-07342-f017:**
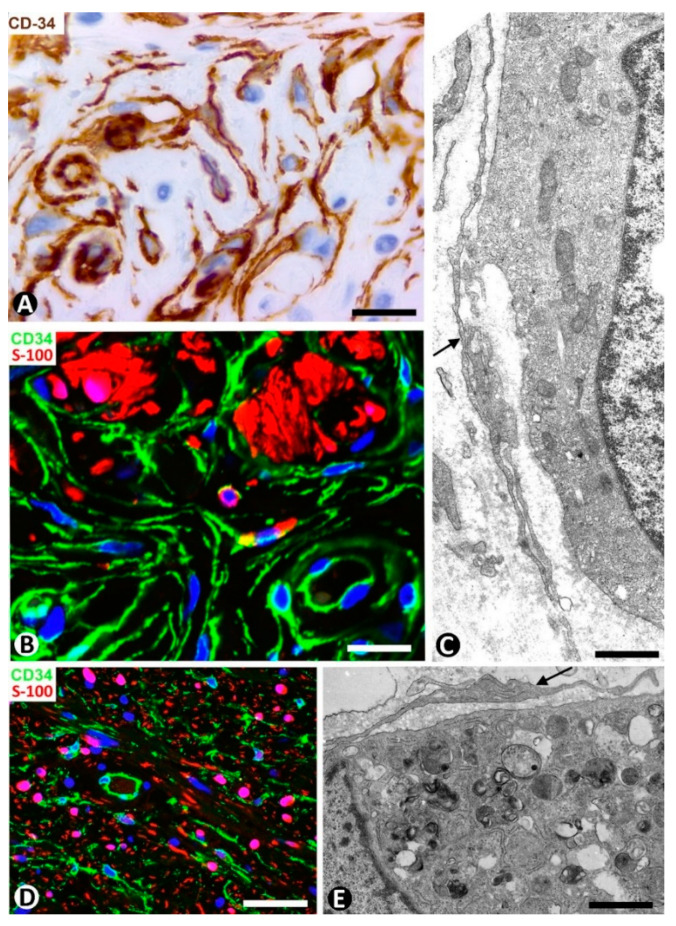
CD34+SCs/TCs in dermal melanocytic nevi (**A**–**C**), neurofibroma (**D**) and granular cell tumor (**E**). (**A**) Immunochemistry for CD34. Hematoxylin counterstain. (**B**,**D**) Double immunofluorescence for CD34 (green) and S100 protein (red). DAPI counterstain. (**C**,**E**) Ultrathin sections. Uranyl acetate and lead citrate. (**A**,**B**) CD34+SCs/TCs (brown in (**A**), green in (**B**) around nevic cells (red in B). (**C**) Ultrastructural image of long, thin telopodes (arrow) around a nevic cell. (**D**): Presence of CD34+ stromal cells (green) and Schwann cells (red) in a neurofibroma. (**E**) Ultrastructural image of a telocyte process (arrow) around a granular cell in a granular cell tumor. Bar: (**A**,**B**) 25 µm, (**C**,**E**) 4 µm, (**D**) 45 µm.

**Table 1 ijms-22-07342-t001:** Contributions about CD34+SCs/TCs in the skin.

In Normal Conditions	In Pathological Conditions
Identification, distribution, morphology, ultrastructure	Scleroderma [38,47,48]
and immune markers [13,35,36,37,38,39,40,41,42,43,44,45]	Psoriasis [49]
Isolation, in vitro studies, cytokine profile [18,43,45]	Wound Healing [45]
Hypothesized roles [3,4,13,14,18,22,35,36,37,38,39,40,41,42,43,44,45]	Basal and squamous cell carcinoma [50]

## Data Availability

All the data are reported in the present paper.

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
