# Peer review of "Cd34+ Stromal Cells/Telocytes in Normal and Pathological Skin"

_ijms, 2021, doi:10.3390/ijms22147342_

Round 1

Reviewer 1 Report

The paper entitled 'CD34+ STROMAL CELLS/TELOCYTES IN NORMAL AND PATHOLOGICAL SKIN' by Diaz-Flores and colleagues is well-written review with multiple original images obtained by immunostaining that describe the normal and pathological conditions. In particular, the staining of stromal cells/telocytes in these tissues is very helpfull, being a comprehensive morphological description of these cells and their alterations related with different skin pathologies.

 I suggest the authors to include some references in their manuscript

Fernandes NRJ, Reilly NS, Schrock DC, Hocking DC, Oakes PW, Fowell DJ. CD4+ T Cell Interstitial Migration Controlled by Fibronectin in the Inflamed Skin. Front Immunol. 2020 Jul 24;11:1501. doi: 10.3389/fimmu.2020.01501. PMID: 32793204; PMCID: PMC7393769.

Gaylo A, Overstreet MG, Fowell DJ. Imaging CD4 T Cell Interstitial Migration in the Inflamed Dermis. J Vis Exp. 2016 Mar 25;(109):e53585. doi: 10.3791/53585. PMID: 27078264; PMCID: PMC4841317.   Wang L, Song D, Wei C, Chen C, Yang Y, Deng X, Gu J. Telocytes inhibited inflammatory factor expression and enhanced cell migration in LPS-induced skin wound healing models in vitro and in vivo. J Transl Med. 2020 Feb 6;18(1):60. doi: 10.1186/s12967-020-02217-y. PMID: 32028987; PMCID: PMC7003342.  

Cretoiu D, Radu BM, Banciu A, Banciu DD, Cretoiu SM. Telocytes heterogeneity: From cellular morphology to functional evidence. Semin Cell Dev Biol. 2017 Apr;64:26-39. doi: 10.1016/j.semcdb.2016.08.023. Epub 2016 Aug 25. PMID: 27569187.

Author Response

The references suggested have been included in new paragraphs and in Table I. For example:

“In the skin, the suggested roles of CD34+SCs/TCs include mechanical support, regeneration (tandem between TCs and stem cells), communication (intercellular contacts and extracellular vesicles) and endocytosis, immune regulation, modulation of fibroblasts, mast cells and macrophages, reduction of inflammatory response, participation in metabolism, homeostasis, (neo) angiogenesis, and in the interaction between collagen and elastic fibres [3, 4, 13, 14, 15, 18, 22, 35-45]. “

“Several factors are known to be involved in immune cell control in the interstitial migration of CD4+ T lymphocytes in the inflamed dermis [79-81] and CD34+SCs/TCs in the reduction of the inflammatory response in skin wound healing models [45]. Future studies are required to typify inflammatory cells arranged around vessels and surrounded by perivascular CD34+SCs/TCs in the reticular dermis affected by pathological processes, and to evaluate the possible role of CD34+SCs/TCs in retaining and modulating inflammatory cells in the perivascular spaces”.

Thank you for improving this manuscript. 

Reviewer 2 Report

CD34+ Stromal Cells/Telocytes in Normal and Pathological Skin

In their paper, Diaz-Flores and colleagues examined telocytes (CD34+ stromal cells) in both normal and pathologic skin, highlighting both their physiological and pathophysiological roles. Furthermore, they properly illustrated their findings in clear immunohistochemistry and electron microscopy images.
The Introduction section offers quite some background information, while clearly stating that the objective of this work is to review the characteristics and morphologic behavior of TCs in both normal and pathological skin. However, it is my opinion that the authors should perhaps also briefly summarize the most important roles of these cells in the skin. Additionally, before further analyzing each area, I suggest the authors add a table highlighting the most important aspects identified in the addressed conditions.
The authors state that ‘Most CD34+ stromal cells are known to correspond to telocytes, a new cellular type identified by electron microscopy’. Since there is some controversy regarding the most adequate way to correctly identify telocytes, especially depending on the anatomical site, the authors should perhaps concisely address this issue, using appropriate references that are lacking in the current form. 
Further on, figures seem free from apparent manipulation and are of high enough quality for the reader to interpret, while legends are clearly explained. The writing is generally very clear and well organized, while the English language and style are of sufficient quality. Apart from some minor mistakes that can be easily addressed during the proofreading phase, I found no issues regarding this aspect.
The meaning of the study’s findings is addressed throughout the paper, focusing on how it relates to the existing literature on the subject. While the conclusion of the manuscript is clear, highlighting the need for future studies in this area, the authors could maybe mention the practical applicability of these findings. 
 Overall, this is a sound manuscript of sufficient interest for the readers. Following minor revision, I believe that it could be published in The International Journal of Molecular Sciences.

Author Response

Regarding the general hypothesized roles of CD34+SCs/TCs, we have briefly mentioned those suggested by other authors for these cells in the skin:

“In the skin, the suggested roles of CD34+SCs/TCs include mechanical support, regeneration (tandem between TCs and stem cells), communication (intercellular contacts and extracellular vesicles) and endocytosis, immune regulation, modulation of fibroblasts, mast cells and macrophages, reduction of inflammatory response, participation in metabolism, homeostasis, (neo) angiogenesis, and in the interaction between collagen and elastic fibres [3, 4, 13, 14, 15, 18, 22, 35-45]”.  

In the introduction, we have added a table (Table I) with the most important aspects identified in these cells.

Regarding the correct identification of telocytes, we have added the following paragraph on their immunophenotypic heterogeneity and the immune markers used.

“Although telocytes show a characteristic ultrastructure, they present immunophenotypic heterogeneity, depending on anatomical location [3]. In addition to CD34 positivity, TCs are also described as CD34/PDGFRα double positive [4-11]. Expression of vimentin, CD117, CD29, VEGF, and estrogen and progesterone receptors has also been observed in some locations [12-17]. The immunophenotype profile of telocytes (CD34+/PDGFRα+/vimentin+/CD31-) differentiates them from fibroblasts (CD34-/PDGFRα+/vimentin+/CD31-) and endothelial cells (CD34+/PDGFRα-/vimentin+/CD31+) [18]. In general, a specific immune marker for TCs is still an issue of study”.

In the conclusion, we mention the practical applicability of the findings as an issue for future studies. In addition, in section 5 (considerations and future studies….) we have included the following:

“Future studies should also consider the practical applicability of the findings described for CD34+SCs/TCs in the normal and pathologic skin, including their possible therapeutic implications.”.

Thank you for improving this manuscript.